# Delayed Antarctic sea-ice decline in high-resolution climate change simulations

Thomas Rackow [1,5✉], Sergey Danilov [1,2], Helge F. Goessling [1], Hartmut H. Hellmer [1], Dmitry V. Sein [1,3], Tido Semmler [1], Dmitry Sidorenko [1] & Thomas Jung [1,4]

Despite global warming and Arctic sea-ice loss, on average the Antarctic sea-ice extent has not declined since 1979 when satellite data became available. In contrast, climate model simulations tend to exhibit strong negative sea-ice trends for the same period. This Antarctic sea-ice paradox leads to low confidence in 21st-century sea-ice projections. Here we present multi-resolution climate change projections that account for Southern Ocean mesoscale eddies. The high-resolution configuration simulates stable September Antarctic sea-ice extent that is not projected to decline until the mid-21st century. We argue that one reason for this finding is a more realistic ocean circulation that increases the equatorward heat transport response to global warming. As a result, the ocean becomes more efficient at moderating the anthropogenic warming around Antarctica and hence at delaying sea-ice decline. Our study suggests that explicitly simulating Southern Ocean eddies is necessary for providing Antarctic sea-ice projections with higher confidence.

[1] Alfred Wegener Institute, Helmholtz Centre for Polar and Marine Research, Bremerhaven, Germany. [2] Jacobs University Bremen, Bremen, Germany. [3] P.P. Shirshov Institute of Oceanology, Russian Academy of Sciences, Moscow, Russia. [4] Institute of Environmental Physics, University of Bremen, Bremen, Germany. [5] Present address: European Centre for Medium-Range Weather Forecasts, Bonn, Germany. ✉email: thomas.rackow@awi.de

Climate models that participated in the Coupled Model Intercomparison Project Phase 5 (CMIP5)[1] simulate a strong decline of September Antarctic sea-ice extent (SIE) for recent decades (Supplementary Fig. 1). In contrast, satellite data show a much more stable sea-ice cover on average, with a slight (albeit statistically non-significant) multi-decadal positive trend of +0.11 million km$^2$ per decade[2] in September for the same period (1979–2018). This supports the notion of a possible delayed sea-ice decline in the Antarctic compared to that in the Arctic. In the few years from 2016 onwards, there has been significantly lower Antarctic SIE[3–8], which one could consider harbingers of imminent change. However, others[9] have argued that SIE is expected to regress to the near-neutral decadal trend in the near future; and indeed data for 2020/2021 seem to support this view[8].

To explain the lack of long-term Antarctic sea-ice decline since 1979, previous studies have proposed various mechanisms such as decadal wind trends associated with the Southern Annular Mode (SAM)[10] and variability of the Amundsen Sea Low[11,12], dynamic sea-ice transport changes[13–16], recent absence of deep open-ocean convection[17,18], additional meltwater from the Antarctic ice sheet and ice shelves[19–21], and internal variability of tropical origin[22,23]. While some of these factors are likely to contribute to the slightly positive real-world trend, here we argue that mesoscale dynamical processes in the ocean play a critical role for explaining a delayed Antarctic sea-ice decline. Due to the contrasting trends between observations and existing climate models (Supplementary Fig. 1), in which mesoscale oceanic processes are parameterized, as well as due to a lack of consensus regarding possible explanations for these differences[24], the Intergovernmental Panel on Climate Change (IPCC) concludes that our confidence in projections of Antarctic sea ice is low[24–28].

It has been suggested that models with a faithful representation of mesoscale ocean eddies in the Antarctic Circumpolar Current (ACC) are needed in order to better represent the behaviour of the Southern Ocean[29]. In fact, realistically representing the Southern Ocean circulation and its response to global warming is known to be a challenging task in coarse-resolution models[30], requiring parameterizations (e.g. for the effect of ocean eddies) that are sensitive to how they are tuned. In particular, changes in forcing can be compensated via eddy compensation and saturation[31], resulting in a notably insensitive transport of the ACC through Drake Passage. Both changes in mesoscale transient eddies and changes in meandering flows[32] (standing eddies) may contribute to these phenomena, and the degree to which eddy compensation and saturation is realized in simulations thus depends on parameterizations in low-resolution (LR) models. There is some level of arbitrariness involved due to necessary choices being made for individual implementations of parameterizations in climate models. Although the development of eddy parameterizations is still an active field of research, it is increasingly being recognized that explicitly simulating mesoscale processes is often more accurate[33]. Therefore, an alternative way is to develop and use high-resolution (HR) sea ice-ocean models[34] that employ more of the laws of physics by explicitly simulating rather than parameterizing the effects of ocean eddies. Compared to LR simulations for the Southern Ocean, earlier studies using HR ocean components have reported changes in the hydrography[35] as well as of the overturning and ocean heat transport in idealized climate change simulations[32,35].

Here, we show the impact of explicitly simulating mesoscale processes in the ocean on Antarctic sea-ice trends in comprehensive CMIP-type projections using a high-emission scenario. Our HR climate model configuration simulates stable September Antarctic SIE that is not projected to decline until the mid-twenty-first century. This is in stark contrast to typical CMIP models and our complementary LR configuration. From additional mixed-resolution experiments, in which only the atmosphere or ocean is highly resolved, we attribute this delayed decline to the HR ocean. Our study provides support for the hypothesis that ocean models that permit or resolve eddies are necessary to increase the trustworthiness of climate projections of the Southern Ocean, and it thus constitutes a step towards providing Antarctic sea-ice projections with higher confidence.

## Results

**Resolution dependence of sea-ice trends in climate models.** When considering the CMIP5 ensemble of models, we find evidence that simulated historical (1979–2018) sea-ice trends depend on the resolution of the ocean model used in the Southern Ocean. Even in the low spatial resolution range of 35–200 km available from CMIP5, models with higher resolution tend to show positive or moderate negative SIE trends ≥ −0.2 million km$^2$ per decade; the coarsest models, on the other hand, show negative trends amounting to as much as −0.8 million km$^2$ per decade (Fig. 1b). Particularly, the one CMIP5 model with the highest spatial ocean resolution in the Southern Ocean (35 km), MPI-ESM-MR[36], is able to simulate a positive SIE trend over the historical period. Taking only those CMIP5 models that realistically simulate the observed September Antarctic SIE mean state (Fig. 1a), the negative correlation of −0.63 between simulated SIE trend and ocean resolution becomes significant (slope different from zero at the 5% level; Fig. 1b). Dedicated higher-resolution experiments are needed to test the hypothesis that simulated sea-ice trends could be tied to the ocean resolution.

We perform experiments with an eddy-permitting and locally eddy-resolving global model system (AWI-CM-HR)[37–40] (down to 8 km resolution in some parts of the Southern Ocean, see "Methods") that has been integrated until the end of the twenty-first century, following the HighResMIP protocol[41]. AWI-CM employs a sea ice-ocean model formulated on multi-resolution unstructured meshes[42]. This technique permits ocean eddies over the whole core of the ACC when compared to the local Rossby radius of deformation[43] by concentrating grid points in that area (Fig. 2g, i). Over the ACC, the HR grid is thus similar in resolution to a 1/10° ocean model that is typically termed "eddy-rich" over mid-latitudes, but stops being eddy-resolving in polar regions[43] (Supplementary Figs. 4 and 5). The HighResMIP experimental design involves complementary experiments at lower ocean resolution similar to the ocean resolution used in CMIP5 (AWI-CM-LR, Supplementary Fig. 4 and 5), without re-tuning the model to isolate the sole impact of resolution[44]. To our knowledge, besides more idealized climate change simulations[32,35,45] or relatively short integrations[46,47] at eddy-rich resolutions, comprehensive climate change projections until the end of the century have not been performed yet using an ocean model that is eddy-permitting in the vicinity of the whole ACC core.

**2m temperature and September Antarctic sea-ice extent projections.** Temperature projections for the end of the twenty-first century (2070–2099), under a representative high-emission concentration pathway (RCP8.5, see "Methods"), depict typical global warming patterns with some well-known features (Supplementary Fig. 6). For example, both HR and LR projections show stronger warming over the continents than over the ocean[48]; a cold blob over the North Atlantic[49] is evident at both resolutions that is probably associated with a slowdown of the Atlantic Meridional Overturning Circulation[49–51]; and Arctic Amplification, which is strongest in winter, is clearly discernible[52]. Importantly, large differences between the HR and LR projections

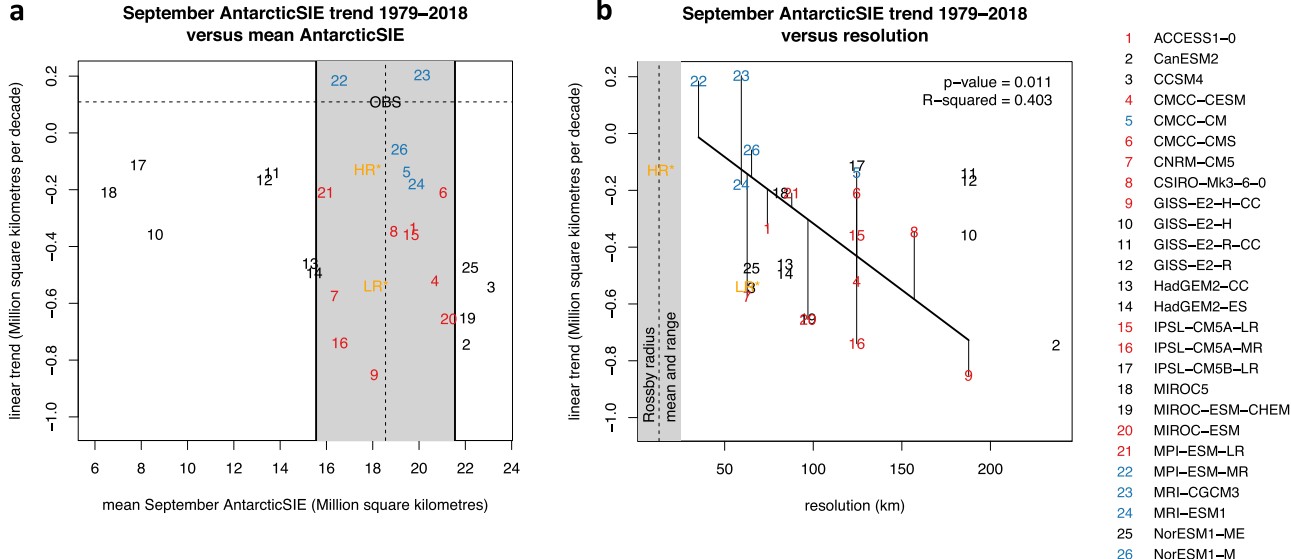

**Fig. 1 September Antarctic sea-ice extent (SIE) trends during the satellite era (1979–2018) in current climate models participating in the Coupled Model Intercomparison Project (CMIP5). a** Trends in SIE as function of mean SIE. The grey box encompasses CMIP5 models that are close to the observed mean September SIE (OBS[97], vertical dashed line) within ±3 million km$^2$. Those models with positive or only moderate negative linear trend (within ±0.3 million km$^2$ per decade of the observed trend, horizontal dashed line) are coloured blue, the other models with stronger negative trends are coloured red. Numbers represent CMIP5 models listed to the right. **b**, Trends in SIE as function of average ocean resolution in the Southern Ocean (45–65° S latitudinal band). The vertical dashed line denotes the mean Rossby radius of deformation in that band (13 km), while the grey box defines its range. For the models with good representation of mean SIE, a linear relationship between trend and ocean resolution is evident, with slope significantly different from zero at the 5% level ($p = 0.011$). The average ocean resolution was computed for each of the CMIP5 models as the square root of the average grid cell area of all ocean model cells located in that band, according to their centre (see "Methods"). Only ensemble means are given for each CMIP5 model (an otherwise identical figure with all ensemble members is provided in Supplementary Figs. 2 and 3). HR* and LR* (orange) are results from the two high- and low-resolution AWI-CM simulations discussed in this paper, but for the later period 2019–2058. High-resolution SIE decrease is delayed compared to the low-resolution configuration (see "Methods" for a discussion of earlier trends, including 1979–2018).

can be seen in the Southern Hemisphere (Fig. 2c, f), where the HR experiment shows a much lower projected near-surface warming. Around the Antarctic continent, the HR temperature response is generally lower by 1–2 °C, and locally reduced by up to 3 °C in the DJF (December–January–February) season and 5 °C in JJA (June–July–August) when compared to LR.

Consistent with the temperature changes, projected September SIE in the Antarctic is different between the two experiments (Fig. 3a): The SIE decline in the coarse-resolution LR experiment (−0.49 million km$^2$ per decade between 1979–2018; Supplementary Fig. 2) is similar to what can be found for the CMIP5 ensemble (Fig. 1), with strong projected sea-ice decrease until the end of the century that already sets in around the 1980s. The HR experiment, on the other hand, exhibits enhanced decadal-scale variability in SIE between 1950 and 1990 (especially in the Weddell Sea sector; Supplementary Fig. 7), which has been found in other HR models[53,54]. This probably spurious behaviour may still be related to the spinup, given that it disappears in the scenario after about the year 2010 (Fig. 3a). The Southern Ocean is weakly stratified[55] and sea ice in HR appears to be more sensitive to initial adjustments than in LR. Interestingly, there is evidence for multi-decadal sea-ice variability from reconstructions[56–58]. These suggest a twentieth century decline of Antarctic SIE, by up to 20% between the 1950s and 1990s[57], which precedes the later increase found in post-1979 satellite data[58]. Importantly, the projected HR SIE shows no clear decline until about the year 2050 (still only about −0.13 million km$^2$ per decade between 2019–2058; Fig. 1), and even afterwards the decline progresses with a reduced rate compared to the LR experiment (Fig. 3). The same holds for the Antarctic sea-ice volume (Fig. 3b), where LR loses a volume of $\Delta V = 5900$ km$^3$ until the end of the century (2070–2099) compared to 1990–2019, while HR loses $\Delta V = 1600$ km$^3$.

Considering the spatial fingerprint of sea-ice change, the sea-ice concentration in HR shows a pan-Antarctic decrease along the ice edge that is much smaller than in LR (Fig. 3c). This annular structure of the difference between HR and LR is consistent with the notion that mesoscale ocean eddies, which are abundant in the ACC surrounding Antarctica, play a critical role in the delayed sea-ice decline. It is important to realize that differences between HR and LR are not solely due to changes in the Weddell Sea, where the model is subject to enhanced decadal variability during its initial adjustment (Eastern Weddell Sea between 0° and 70° E). In fact, when excluding the Weddell Sea area from the analysis, the differences between LR and HR remain (Supplementary Fig. 8) showing that the HR model is not just recovering from an open-ocean Weddell Sea polynya[18]. Furthermore, the weaker pan-Antarctic fingerprint in HR when compared to LR is also clearly visible when using different baselines for comparing the late twenty-first century changes (Supplementary Fig. 9), showing that the differences are independent of the phase of the initial adjustment. As will be shown below, the large annular differences between HR and LR can be understood physically by invoking mesoscale ocean processes in the Southern Ocean.

In order to understand the role of enhanced resolution in the ocean vs that in the atmosphere, a second set of "mixed-resolution" experiments was performed. In these experiments, the HR ocean was run with the LR atmosphere, and vice versa (see "Methods"). The mixed-resolution simulations reveal that the cause for the delayed sea-ice decline is likely to be found in the HR ocean (Supplementary Fig. 10), with no discernible influence from atmospheric resolution, at least for the resolutions considered here. Due to the different atmospheres, these experiments can also be considered as another 'perturbed-physics' ensemble member for each ocean grid.

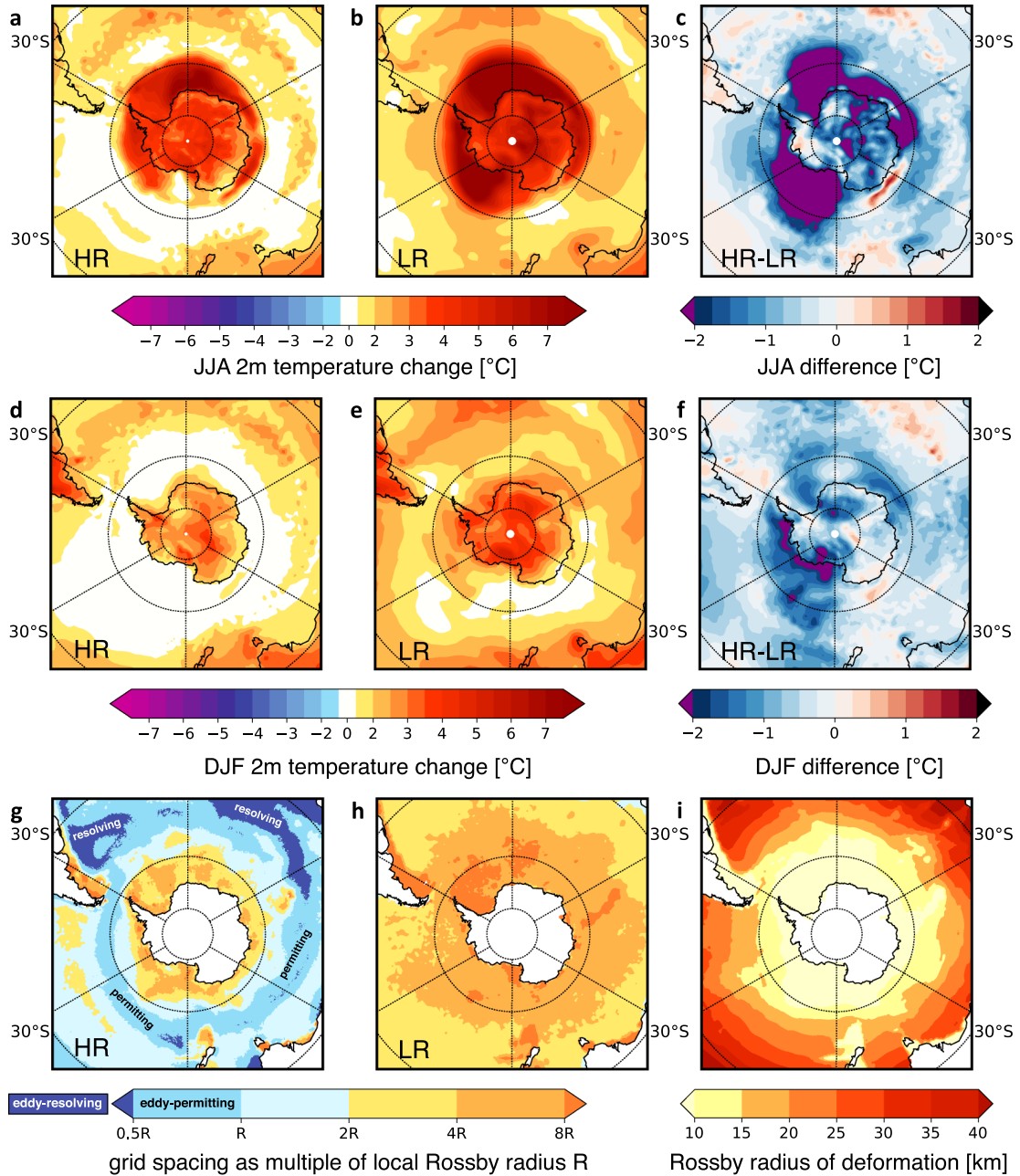

**Fig. 2 Projected 2m temperature changes for the end of the twenty-first century (2070–2099, relative to 1990–2019) in the JJA (June–July–August) and DJF (December–January–February) seasons.** The projections with the **a**, **d** high-resolution (HR) and **b**, **e** low-resolution (LR) model configurations are based on a Representative high-emission Concentration Pathway (RCP8.5 scenario). Differences in the climate response for **c**, **f** the HR and LR projections (HR-LR) are most pronounced in the Southern Hemisphere. A "delta approach", as outlined by the HighResMIP protocol (High-Resolution Model Intercomparison Project, see "Methods"), has been applied to isolate the climate change signal. The ocean grid spacing for **g** HR and **h** LR is given as a multiple of the local **i**, Rossby radius $R$ of deformation ("Methods").

In our model simulations, the projected increase in the strength of the westerlies around Antarctica (Supplementary Fig. 11), which is driven by ozone depletion and greenhouse gas forcing[59], is about 5–7%. This value is not sensitive to the atmospheric resolution, which further shows that the atmosphere is not the main source of the differences seen in our simulations. In summary, our results suggest that the use of coarse-resolution ocean components in current climate models, which require parametrization of the effects of mesoscale ocean eddies, contributes to the fact that those models often simulate an early sea-ice decline that is inconsistent with observations. Next, we propose a possible physical explanation for this finding and

discuss important differences in ocean heat transport between the LR and HR simulations.

**Simulated Southern Ocean mean state**. Two notable differences between the high- and low-resolution experiments in the Southern Ocean are (i) the water mass structure and (ii) the resulting meridional overturning circulation. In the following, both aspects will be discussed. Starting with the water mass properties, LR shows a pool of warm subsurface waters southward of 60° S, which is up to 1 °C warmer than observed in the upper 500 m (Fig. 4d, f). The HR experiment, on the other hand, maintains a sharper meridional temperature gradient and simu-

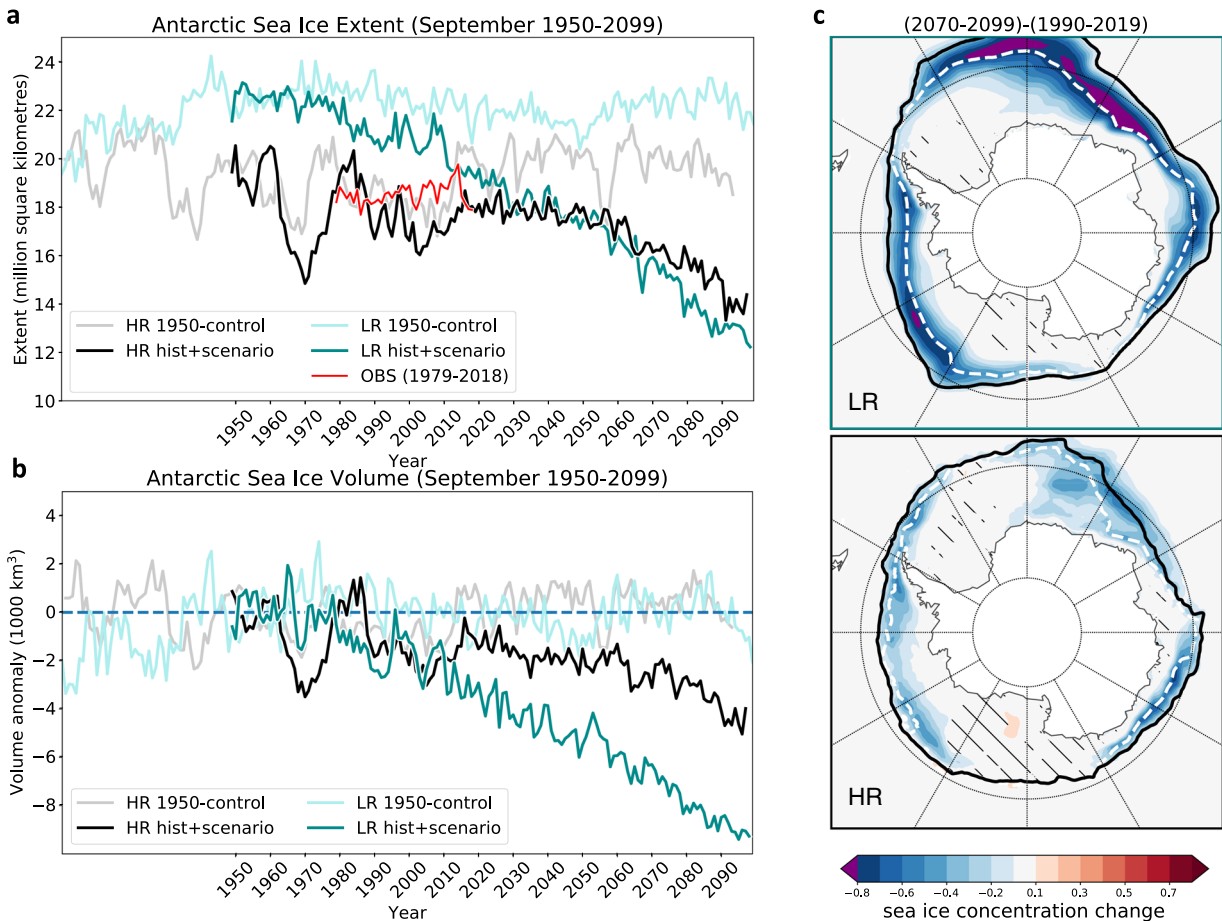

**Fig. 3 Time series of observed historical September sea ice (1979–2018) and projections until the end of the twenty-first century. a** Historical and scenario simulations of September Antarctic sea-ice extent and **b** sea-ice volume anomaly in the high-resolution (HR; grey/black) and low-resolution (LR; light green/green) AWI-CM configurations. Control simulations with atmospheric greenhouse gas concentrations fixed at 1950 levels are shown in light colours; the red line depicts historical SIE observations (OBS) over the satellite era (see "Methods"). The volume anomalies are relative to 1950-control means. **c** Patterns of projected sea-ice concentration changes in September (2070–2099 minus 1990–2019) in LR and HR. Black (white dashed) contours show the sea-ice extent for 1990–2019 (2070–2099). The "delta approach" (see "Methods") has been applied in (**c**) to isolate the climate change signal. The signals are significantly larger than typical unforced 30yr-variability in the 1950 control simulations (see "Methods"). Hatching indicates non-significant regions where the concentration changes are smaller than two standard deviations.

lates a cooler subsurface compared to LR (~0.5 °C cooler than PHC), effectively making the sea ice less vulnerable to vertical entrainment of heat. The erroneously warm subsurface waters in LR are largely of Atlantic origin from a depth of about 2000 m, as shown by relevant $\sigma_2$-density isopycnals (isolines of potential density, referenced to 2000 m depth). The upwelled water of Atlantic origin ($\sigma_2 \approx 37.1$) extends too far south in the low-resolution experiment, probably in part due to numerical diapycnal mixing, leading to the relatively warm and salty subsurface conditions. Another contribution could be the combined effect of the Gent–McWilliams (GM) parameterization[60] for mesoscale eddies, which acts to flatten isopycnals, and isoneutral (Redi) diffusivity. This latter mixing along isopycnals is not well constrained and might lead to diffusive transport of warmer waters. A similar statement as for the upwelled waters holds for lighter waters ($\sigma_2 \approx 36.1$–36.4) that reach under the sea ice in LR (white contours in Fig. 4c), which is in contrast to the observed outcropping northwards of the sea-ice edge in the observations and in HR (Fig. 4a, b). Again, HR is able to maintain a sharper density contrast between the water masses of Atlantic and Antarctic origin in the 1000–2000 m range.

The canonical pattern of the overturning circulation in the Southern Ocean is known to be the result of the balance between

the wind-driven and eddy-induced overturning cells[61]. This residual circulation is difficult to estimate directly from observations[62]; therefore its magnitude is still rather uncertain. Estimates also vary considerably across ocean models[63]. Using salinity as a tracer, observed present-day salinity sections indirectly reveal the structure of the residual circulation (e.g., Fig. 1 by Armour et al.[64]), with a marked upwelling branch of saline waters within the region 40–70° S (Fig. 4a). Part of this upwelled water flows to the south and transforms into the densest water of the World Ocean, Antarctic Bottom Water, while the other part flows northward, where it mixes with freshwater from sea-ice melting and interacts with the atmosphere, and gradually subducts northwards of 50° S as Antarctic Intermediate Water and Subantarctic Mode Water[65].

Salinity sections along 30° W indirectly illustrate that the structure of the simulated residual overturning circulation in the HR experiment is more realistic compared to LR (Fig. 4b, c), with a stronger northward branch of the upper circulation cell as indicated by the prominent salinity minimum layer north of the ACC (and associated stronger northward heat transport in 1990–2019, see below). Compared to observations, HR also shows a more realistic salinity distribution in the region 50–80° S, which implies a better representation of the upwelling of relatively saline

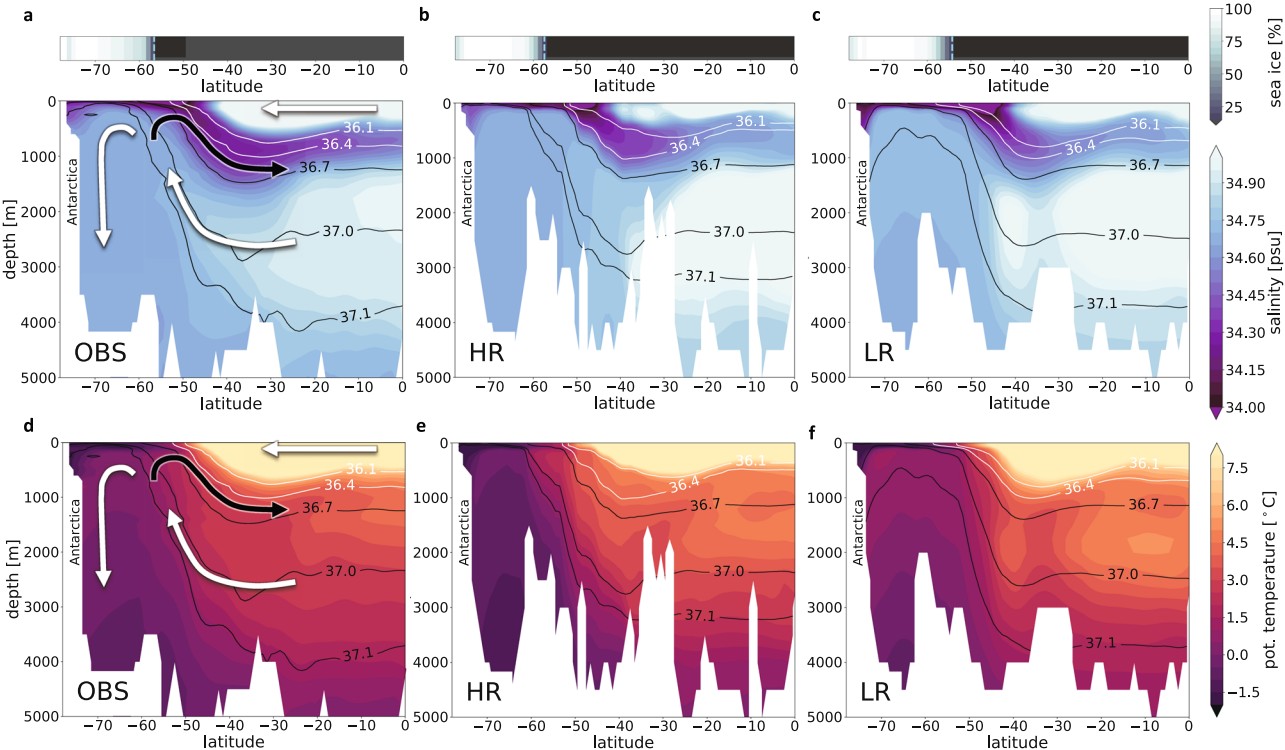

**Fig. 4 Southern Ocean sea-ice fraction [%], salinity [psu], and potential temperature [°C] sections along the 30° W meridian in present-day observations and in model simulations (1990–2019). a, d** Observations (OBS) from the National Snow and Ice Data Centre NSIDC[98] (September sea-ice fraction) and from the Polar science centre Hydrographic Climatology PHC 3.0[92] (salinity and potential temperature), overlaid with a schematic representation of the residual overturning circulation (arrows), **b, e** simulated water mass structure and sea-ice fraction for the high-resolution configuration (HR), and **c, f** same for the low-resolution (LR) simulation. Dashed vertical lines denote the latitude of September sea-ice edges. For better orientation between the different panels, relevant density contours ($\sigma_2$-isopycnals) are highlighted (black and white lines). Black arrows depict the northward branch of the upper circulation cell of the residual overturning circulation that is connected to the prominent low-salinity layer of Antarctic Intermediate Water.

waters and supports a more realistic formation of deep and bottom waters, as was shown in previous simulations using the HR ocean grid[66].

**Meridional ocean heat transport and climate change.** We argue that the different functioning of explicitly simulated versus parameterized eddies lies at the core of the different sea ice-ocean response in our simulations. One aspect of this response is illustrated in detail below in terms of the behaviour of meridional ocean heat transport (MHT). Observations and previous ocean simulations[64] suggest that equatorward ocean heat transport and circumpolar upwelling has delayed warming in the Southern Ocean over the satellite era. In the following, we explore the link between ocean heat transport changes, different magnitudes of projected Southern Ocean warming, and the development of Antarctic sea ice during the twenty-first century.

The shape of the MHT is largely determined by the vertical structure of the ocean circulation[67] in the upper ~1000 m. The canonical MHT is generally poleward (negative values in Fig. 5a); and the poleward MHT between 45° S and 60° S across the ACC is larger in LR when compared to HR. This result is consistent with earlier findings[32]. It may seem counterintuitive at first, given that a HR model can actually permit or even resolve mesoscale eddies, which generally transport heat polewards[68,69] across the ACC. However, the relative magnitude of eddy heat transport in a HR compared to a low-resolution model depends on the details of the eddy parameterization applied in the low-resolution model (e.g., a higher eddy coefficient chosen in the GM parameterization[60] for mesoscale eddies can induce larger eddy-induced transports[70]).

The heat transport associated with the northward residual branch of the upper circulation cell (black arrow in Fig. 4a), made visible as a prominent salinity minimum layer, causes a reduction of the poleward heat transport around 45° S. In a changing climate, this upper cell of the residual circulation appears to be more efficient in moderating warming in the upper Southern Ocean in the HR experiment (Fig. 5c), as evidenced by an absolute equatorward heat transport in the vicinity of 45° S for 2070–2099 (Fig. 5a). Between 40 and 80° S, HR consistently simulates positive (equatorward) MHT changes at the end of the twenty-first century (Fig. 5d). The surface and upper subsurface—especially along the sea-ice edge—thus remains considerably cooler than in LR (compare Fig. 5e, f), as clearly seen in projected near-surface temperature changes northward of the ice edge at roughly 60° S (compare Fig. 2c, f).

In a previous modelling study by Bitz and Polvani[71] with the CCSM3.5 model, a resolution-dependent additional warming in their 1° case was found mostly northward of about 60° S, away from the ice edge. This finding has been used by the authors as a possible explanation for the more similar sea-ice response (for different ocean resolutions) than might have been anticipated. According to the concept of eddy saturation[31], ocean eddies compensate for changes in external forcing. Both transient and standing eddies may contribute to this phenomenon, and the simulated compensation is dependent on parameterizations in low-resolution models. In contrast to HR, our low-resolution climate change experiment shows increased poleward MHT even south of 60° S (Fig. 5d). Again, this is consistent with an earlier finding with the CCSM3.5 model[32] where parameterized

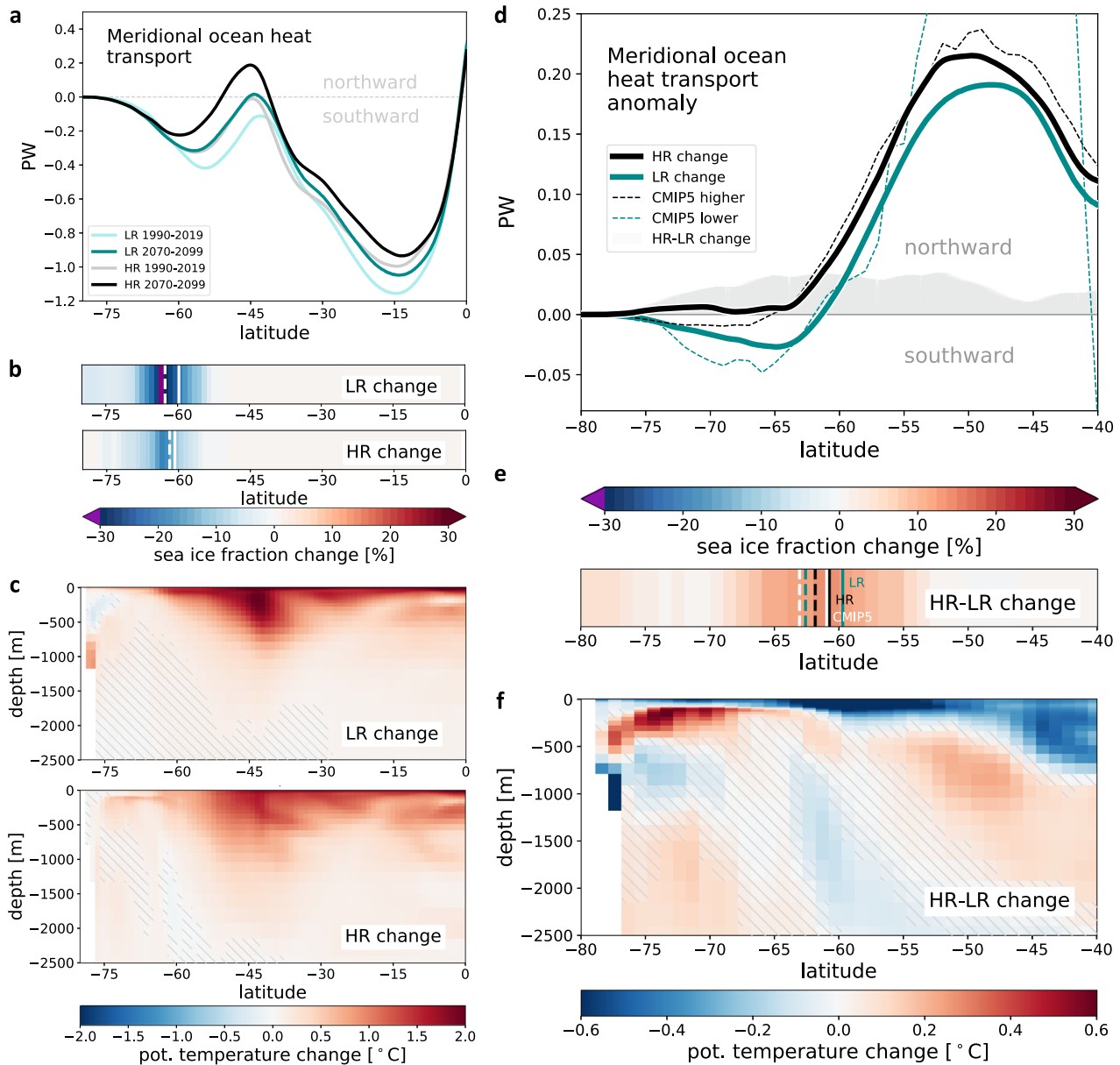

**Fig. 5 Meridional ocean heat transport MHT [1 PW = 10$^{15}$W] in the Southern Hemisphere and its link to zonal-mean potential temperature [°C] and zonal-mean sea-ice changes [%]. a** MHT in the low-resolution (LR) and high-resolution (HR) experiment for 1990–2019 (light green/grey) and 2070–2099 (dark green/black). **b** Zonal-mean sea-ice change and **c** zonal-mean ocean potential temperature change in LR and HR between 1990–2019 and 2070–2099. Vertical solid lines in panel (**b**) give the location of the September sea-ice edge in 1990–2019, vertical dashed lines the location in 2070–2099. **d** Zoom (40–80° S) into the anomalous MHT (2070–2099 minus 1990–2019). Climate models participating in the Coupled Model Intercomparison Project (CMIP5) are grouped into two sets with 'higher'/'lower' resolution than 90 km (for the computation, see "Methods"). **e** Difference in the response (HR-LR) of the panels in (**b**). Vertical solid (dashed) lines denote the location of the September sea-ice edge in 1990–2019 (2070–2099) in LR, HR, and in CMIP5 ("Methods"). **f** Difference in the response (HR-LR) of the panels in (**c**). The signals are significantly larger than typical unforced running 30yr-variability (standard deviation) of zonal-mean potential temperature in the 1950 control simulations. Hatching indicates non-significant regions where the signals and difference in the response are smaller than two standard deviations (see "Methods"). The "delta approach" (see "Methods") has been applied in (**b**)–(**d**).

poleward eddy heat fluxes increased much more than the explicitly simulated eddy heat flux after a doubling of $CO_2$ (Fig. 9a in Bryan et al.[32]). Bryan et al.[32] explain this difference with increased (approximately doubled) heat transport by standing eddies, the meanders of the ACC, which are more pronounced in the HR configuration. In order to compensate for the change in forcing, the heat flux response by transient eddies can thus be much smaller in HR than in LR (where only transient eddy effects are parameterized), despite similar increases in

westerly winds. We suspect that a similar explanation holds for our simulations. Across 60° S, in particular, anomalous equatorward MHT in our HR is larger than in LR by approx. 0.033 PW (1 PW = 10$^{15}$ W) (Fig. 5d). Again, this is in line with the decreased southward eddy heat transport reported in the HR CCSM3.5 model for a $2 \times CO_2$ experiment[32] (+0.03 PW), which the authors find "a little surprising, given that the zonal wind stress does increase slightly". The difference in MHT is in principle sufficient to help explaining the delayed onset and

reduced rate of September sea-ice decline in our HR experiment (see "Methods").

## Discussion

In this study, we show that a HR climate model is capable of simulating a stable Antarctic sea-ice cover over the past few decades in CMIP-type simulations—consistent with available satellite data. This is in stark contrast to the behaviour of relatively coarse-resolution climate models, including those which participated in CMIP5 and CMIP6 in support of IPCC Assessment Reports. In our HR simulations, the decline of Antarctic sea ice is not projected to start before well into the twenty-first century; and this "delayed decline" of the Antarctic sea-ice cover can be traced back to the use of a HR ocean component. As outlined below, the exact reason for this behaviour remains to be determined. However, the faithful representation of ocean eddies by the laws of physics rather than physical parametrizations seems to be one plausible major contributor.

The impact of ocean resolution has been discussed in previous modelling studies[32,33,40,66,68–72] and some of the results, in particular the sign of the MHT response, may seem contradictory to what is concluded here. However, as detailed below, a meaningful comparison with previous studies is surprisingly difficult to carry out given major differences in model forcing (e.g., abrupt change in wind forcing of at least 20%[68,69] vs. slow change in temperature and winds due to increasing greenhouse gas concentrations[32,72]), model tuning (e.g., strength of the eddy parameterization in low-resolution setups[40,70]), scientific question (e.g., the isolated impact of ozone depletion[71] or the impact of resolution on the mean state[33,40,66] vs. response to forcing[32]), and possibly various mechanisms at play[32,70]. Furthermore, what is actually known about the impact of resolution is obtained from a rather limited number of coupled models (namely, e.g., CCSM[32,71]), reflecting the fact that coupled climate modelling with eddy-resolving ocean components is still in its infancy.

It is important to realize that regular-grid models that are termed "eddy-resolving" based on their characteristics in mid-latitudes, e.g., at 1/10° resolution, stop being eddy-resolving in polar regions due to the latitudinal dependence of the Rossby radius of deformation (Supplementary Fig. 5). The same holds for "eddy-permitting" configurations like the widely used ORCA025[54] at 0.25° resolution, which is eddy-permitting only over parts of the ACC (Supplementary Fig. 5). This can coincide with a very weak ACC transport through Drake Passage of about 90 Sv and was shown to improve at either lower or higher resolution[54]. Similar to a 1/10° model at high latitudes, the multi-resolution AWI-CM-HR configuration discussed in this paper is by construction eddy-permitting over the entirety of the ACC core (Supplementary Fig. 5). In terms of barotropic transport around Antarctica (Supplementary Fig. 12), AWI-CM-HR matches the recent observational Drake Passage transport range of 173 ± 11 Sv[54,73] well (178 Sv), while AWI-CM-LR with its 133 Sv shows lower transport, which is well within the CMIP5 range of 155 ± 55 Sv[74]. Both models thus do not suffer from a weak ACC, which is consistent with the fact that they are either lower resolved (AWI-CM-LR with eddy parameterization) or more highly resolved (AWI-CM-HR without paramerization) in that region than typical 0.25° configurations. Fully eddy-resolving models, with local resolutions better than 3–4 km over the ACC, are currently under development, and it remains to be seen how these models will perform in the Southern Ocean.

One possible reason for the resolution dependence of the Antarctic sea-ice cover lies in the different response of the oceanic meridional heat transport (MHT) in AWI-CM for HR compared to LR. This may be due to differences in the representation of the underlying eddy-induced circulation and the degree to which it can oppose or compensate the changes in the wind-driven overturning. A wind-driven increase of the upper clockwise meridional circulation cell, with the northward branch being its upper representation, may be partly balanced by a compensating counter-clockwise eddy overturning in LR, similar to partial (parameterized) eddy compensation as shown for CMIP5 models[75]. In HR, the compensating counter-clockwise eddy overturning seems to increase more slowly than the wind-driven part, although this will need to be shown more directly in future studies. In line with this, the projected anomalous MHT for HR associated with the upper circulation cell shows an increased northward heat transport across 60° S at the end of the century when compared to LR (Fig. 5d). This is despite similar increases of 7% and 5% in peak zonal-mean wind stress in HR and LR, respectively (Supplementary Fig. 11). Because of the lack of centennial projections at HR[76], it has not been directly shown yet how eddy compensation will evolve in HR climate projections, i.e. over longer time scales that go beyond the historical period. Our results thus provide further insights into this issue; and they suggest a role for eddy compensation for explaining structural uncertainties associated with Antarctic sea-ice projections and the Antarctic sea-ice paradox.

It can be argued that either dedicated tuning or new flow-dependent approaches for how the space and time-dependent GM coefficient is determined in modern eddy parameterizations[32] of climate models with lower-resolution ocean grids could overcome some of the shortcomings discussed in this study. However, current eddy parameterizations, with their individual implementation in climate models, often can not account for the full effect of eddies or may generate ambiguous results[77,78]. In fact, while Southern Ocean eddy compensation in a recent coarse-resolution climate simulation[77] was shown to cancel the warming within the temperature inversion layer that had been predicted in an earlier study[78] (as a response to an increase of the SAM index), the authors mention that their finding is in contrast with Bitz and Polvani[71], whose 1/10° coupled model warmed in an ozone-depletion experiment. In their studies with the same coupled model CCSM3.5, Bryan et al.[32] also highlight a role for standing eddies, i.e., meanders of the ACC, in balancing the wind-driven overturning. This mechanism of ACC equilibration will need further investigation in future studies, as its effects are fully beyond the scope of what eddy parameterizations in current climate models can achieve.

More generally, there are other possible mechanisms that could explain part of the resolution dependence of Antarctic sea-ice cover, including those that relate to vertical eddy heat fluxes[33,79] or differing mean states. It has been suggested, for example, that thermodynamic processes may be critical: Thicker sea ice decays faster than thinner ice, which is a basic thermodynamic property of sea ice[32,80,81]. At least for our simulations with AWI-CM, this mean state dependence is not the main explanation for the different sea-ice behaviour (see "Methods"). An open question is why the different mean states arise in the first place. Since the SIE for HR is closer to the observed extent than LR without particular tuning, it is possible that the different sea-ice mean state in LR itself is affected by details of the eddy parameterization and the choice of GM coefficient, which has been shown to determine the pattern of warming south of the ACC[70].

Due to computational constraints in HR climate modelling, the spinup outlined in the HighResMIP protocol is relatively short (see "Methods"). However, we do not think that this has a major impact on our key findings. This is due to the fact that (i) the differences between LR and HR in sea-ice evolution are large (i.e., high signal-to-noise ratio when compared to estimated internal variability in the 1950 control simulations); (ii) differences between HR and LR can be explained physically (i.e., annular

structure of the response to resolution vs. localized variability associated with the spinup); and (iii) the findings are "reproducible" in full vs. mixed-resolution experiments. Having said this, considering a revised version of the HighResMIP protocol to allow for longer spin-ups may be worthwhile for future efforts, especially if faithful simulations of the Southern Ocean are required.

In our study, we also considered results from the wide range of CMIP5 models, which are generally rather coarse, and found additional evidence for a resolution dependence of the timing of Antarctic sea-ice decline (Fig. 1). When it comes to changes in MHT, the lower-resolved CMIP5 models show a pronounced southward MHT increase south of 65° S, whereas the relatively higher-resolved CMIP5 models show a weaker increase (between AWI-CM-HR and AWI-CM-LR, Fig. 5d), consistent with our proposed mechanism. During the time of writing an increasing number of CMIP6 models became available. An early high-level assessment[82] of these CMIP6 models shows the same structural uncertainties of Antarctic sea-ice cover over the satellite era as found for CMIP5. This is consistent with our results, given that none of the CMIP6 models is eddy-permitting over the entire ACC in the Southern Ocean or even eddy resolving.

Our results suggest that HR climate models, which represent mesoscale dynamics of the Southern Ocean by the laws of physics instead of parametrizations, provide a promising alternative[34] to achieve a more faithful representation of the circulation and Southern Ocean hydrography (Fig. 4) and hence projections of their changes in a warming world. This alternative is becoming increasingly feasible[44] through advances in high-performance computing and scalable next-generation climate models[40,83–85]. The implications of this go beyond the physical climate system as "measured" by the Antarctic sea-ice cover. In fact, the upwelled waters around Antarctica still have considerable nutrient concentrations, and sustain up to 40% of the low-latitude primary production[86,87]. Future changes in the amount of carbon and nutrients being transported along this pathway will impact global productivity and air-sea $CO_2$ flux substantially[87,88]. A reduced warming in high vs low-resolution models at depth could also have strong implications for current efforts to estimate the basal melting of Antarctic ice shelves[89] and for projected Antarctic Ice Sheet behaviour in models. A better representation of the Southern Ocean can thus have major implications for biogeochemistry and the global carbon cycle, for the uptake of anthropogenic heat by the ocean and hence global mean temperature change, and for the trustworthiness of representing basal melting of Antarctic ice shelves[89] and in consequence global sea level rise[90].

## Methods

**CMIP5 analyses of sea-ice extent, volume, and ocean heat flux**. To study the resolution dependence in climate models more generally and to put our results into perspective, we used data from the Coupled Model Intercomparison Project Phase 5 (CMIP5[1]). More specifically, we used historical and RCP8.5 scenario simulation data available on the German Climate Computing Centre's (DKRZ) supercomputer Mistral, which is mirrored from the DKRZ node of the Earth System Grid Federation (ESGF, https://esgf.llnl.gov/). For models with multiple ensemble members available for both the historical and the scenario period, we computed ensemble means prior to any further analyses (black symbols in Fig. 1). Only in Supplementary Figs. 2 and 3 we also show individual ensemble members as grey symbols.

For the analysis in Fig. 1, September Antarctic SIE trends were computed for the satellite era (1979–2018). The width of the grey box (±3 million km²) in Fig. 1a was defined subjectively to select only CMIP5 models with a realistic SIE mean state for subsequently determining the trend dependence on spatial resolution in Fig. 1b. With this choice, the relation is statistically significant ($p = 0.011$; $R^2 = 0.403$). The actual choice of the box width turns out not to be critical, given that significant correlations were also found for wider and smaller widths of ±2 million km² ($p = 0.02$; $R^2 = 0.563$) and ±4 million km² ($p = 0.007$; $R^2 = 0.34$), respectively. When including all models, that is, also those with more unrealistic SIE of less than half the observed value, the relation is significant at the 10% level only ($p = 0.098$; $R^2 = 0.11$).

To compute the average ocean resolution in the 45–65° S latitudinal band, for every model we located all ocean model cells in that band according to their centre. The average resolution was then determined as the square root of the average grid cell area.

Data points for AWI-CM at LR and HR (about 63 km and 14 km average resolution in the 45–65° S latitudinal band) are also added to the CMIP5 results in Fig. 1, denoted as LR*/HR*. For the LR and HR models, we give 40yr trends for the later period 2019–2058, for which LR and HR show a similar sea-ice mean state, and which is unaffected by the HR model's initial adjustment. Even for this later period, HR still remains in the group of "blue" models (i.e., a trend close to OBS for 1979–2018), while LR joins the "red" models with stronger negative trends for all considered periods. In Supplementary Figs. 2 and 3, trends for LR and HR are also given for the 40yr-long satellite era 1979–2018 and for ±10yr-shifted 40yr periods. The 10yr-shifted HR trends are within the group of "blue" models and frame the "outlier" HR trend during the satellite era, which is −0.53 million km² per decade (this happens by chance due to multi-decadal variability associated with the initial adjustment in HR), similar to that for LR (−0.49 million km² per decade).

In order to diagnose the location of the sea-ice edge in the AWI-CM simulations (Figs. 4 and 5b, e), in the NSIDC climatology (Fig. 4), and in CMIP5 (Fig. 5e), we first regridded the September sea-ice concentration fields to a 1° longitude-latitude grid and averaged over the respective time periods. Second, for each longitude, we determined the northernmost ice-edge location (where sea-ice concentration equals 15%) using linear interpolation. For the zonally averaged ice-edge locations (Fig. 5b,e) we averaged the latitudes of the ice-edge location rather than averaging the sea-ice concentration fields before deriving the 15% sea-ice concentration in order to avoid a northward-biased ice-edge location.

For the CMIP5 MHT analysis (Fig. 5d and Supplementary Fig. 13), we implemented the "Zigzag" Method[91] (their Section 2) to derive global northward heat transport from the CMIP5 heat flux variables "hfx" and "hfy" in x- and y-direction, thereby accounting for peculiarities of the individual native model grids. The variables "hfx" and "hfy" were available only for a subset of the models listed in Fig. 1, namely ACCESS1-0, CMCC-CESM, CMCC-CM, CNRM-CM5, GISS-E2-R, IPSL-CM5A-LR, IPSL-CM5A-MR, IPSL-CM5B-LR, MPI-ESM-LR, MPI-ESM-MR, MRI-CGCM3, NorESM1-M, NorESM1-ME. Grouping these CMIP5 models based on a spatial ocean resolution criterion (resolution in the 45–65° S latitudinal band finer or coarser than 90 km) splits the models into two sets of similar size with 7 and 6 members, respectively.

**Experimental setup**. Several control and scenario experiments with the coupled AWI Climate Model (AWI-CM)[37,38,40] covering 150-yr periods were performed using a slightly modified version of the HighResMIP protocol[41] (see below). One set of simulations (AWI-CM-HR) was run at eddy-permitting ocean resolution $r$ over the ACC region ($r < R$, where $R$ is the Rossby radius of deformation), with some further eddy-resolving refinements ($r < 0.5R$, using the criterion by Hallberg[43]) to about 8 km over the Agulhas and Brazil-Malvinas Confluence regions (see map in Fig. 4b, c by Sein et al.[39]). For Fig. 2i, the Rossby radius $R$ was computed from the PHC climatology[92] as $R = (\pi|f|)^{-1} \int_{-H}^{0} N(z)\,dz$, where $f$ is the Coriolis parameter, $H$ is the ocean floor, and $N$ is the Brunt-Väisälä frequency. The average spatial ocean resolution in the 45–65° S latitude band amounts to about 14 km, and no eddy parameterization was used in this band (it is generally switched off locally where resolution is higher than 25 km[42]). HR uses a higher-resolution T127 atmosphere (ECHAM6[93]), corresponding to a resolution of about 1°. These simulations are compared to corresponding low-resolution simulations (AWI-CM-LR) to determine the added value of increased resolution. LR uses a nominal 1° ocean mesh (Fig. 4a by Sein et al.[39]) with average spatial ocean resolution of 63 km in the 45–65° S latitude band ($r > 2R$). The influence of eddies is therefore fully parameterized. The LR ocean is coupled to a T63 atmosphere, corresponding to a spatial resolution of ≈2°. In order to better understand the origin of the resolution dependence, another set of "mixed-resolution" simulations was carried out comprising a HR ocean and low-resolution atmosphere, and vice versa. Because of the length of the control and scenario simulations, only monthly output was stored for all experiments. Added meltwater or melt scenarios/projections were not accounted for in these simulations, which could provide further means for delaying Antarctic sea-ice decline[19–21].

**Implementation of the HighResMIP protocol**. Initially, a 50-yr coupled spinup (atmosphere-ocean) with fixed 1950 atmospheric conditions was performed with both AWI-CM configurations, following an uncoupled 5-yr ocean spinup initialized from mean 1950–1954 EN4 ocean reanalysis[94] fields (EN4.1.1, downloaded in May/June 2016). Regarding the spinup of the coupled systems we followed the original HighResMIP protocol[41]. That is, starting from the initial 50-yr spinup, control simulations with fixed 1950 atmospheric conditions were continued for another 150 years for the HR and LR configurations. In parallel, historical and RCP8.5 scenario simulations of same length have been performed in both configurations for the period 1950–2099. CMIP5 forcing has been used since CMIP6 forcing was not available at the time when the model experiments were carried out. For all projections shown (2m temperature in Fig. 2 and sea-ice concentration in Fig. 3c) and for the anomalous MHT and zonal-mean sea ice and potential temperature changes of the 3-dimensional ocean (Fig. 5b–d), potential model drift as estimated from corresponding periods of the "1950"-control simulations has been

subtracted from the projections as suggested by the HighResMIP protocol[41]. This "delta approach" also effectively subtracts 1950-committed warming and only leaves the climate change signal that results from increased greenhouse gas concentrations compared to 1950 levels.

**Sea ice and ocean potential temperature changes compared to unforced variability in the controls**. We estimate unforced variability in the controls to assess the physical significance of the simulated sea-ice concentration and ocean potential temperature changes. To this end, the standard deviation of rolling 30yr-means of sea-ice concentration from the last 80 years of the 1950 control simulations with AWI-CM-LR and AWI-CM-HR are computed (Supplementary Fig. 14). This can be considered only a rough estimate of unforced variability given the limited length of the period available. For ocean potential temperature, we first compute rolling 30yr-means from the same years of the 1950 control simulations with AWI-CM-LR and AWI-CM-HR. In a second step, zonal means are computed for each 30yr-mean, and then the standard deviation is taken over time to estimate unforced variability of zonal-mean potential temperature. For Fig. 5f, we average the internal variability in the LR and HR controls for the estimate of internal variability. The differences of the HR and LR response (HR-LR change), in particular at the surface and subsurface, are larger than the estimated internal variability (two standard deviations).

**Meridional heat transport and sea-ice changes**. Changes in the amplitude of the MHT can be related to the simulated sea-ice changes as follows. The energy $\Delta E$ needed to melt a volume $\Delta V$ [$m^3$] of sea ice is $\Delta E = \rho_{ice} \times L_f \times \Delta V$, where $L_f = 3.34 \times 10^5$ J/kg is the latent heat of fusion, and $\rho_{ice} = 910$ kg/m$^3$ is a typical density of sea ice as set in the ocean model. The reduction of ice volume between 1990–2019 and 2070–2099 in LR ($\Delta V_{LR} = 5.9 \times 1000$ km$^3 = 5.9 \times 10^{12}$ m$^3$) thus requires $\Delta E_{LR} = \rho_{ice} \times L_f \times \Delta V_{LR} = 1.79 \times 10^{21}$ J. The according smaller change in HR ($\Delta V_{HR} = 1.6 \times 10^{12}$ m$^3$) requires only $\Delta E_{HR} = \rho_{ice} \times L_f \times \Delta V_{HR} = 0.49 \times 10^{21}$ J. The change in MHT across 60° S until the end of the century is larger in HR than in LR by 0.033 PW = $0.033 \times 10^{15}$ W = $3.3 \times 10^{13}$ J/s (Fig. 5d). Between 1990–2019 and 2070–2099, this amounts to an additional northward transport of about $4.2 \times 10^{22}$ J out of the Southern Ocean in HR compared to LR. Despite the seemingly subtle changes in MHT, this is much more than the difference in heat needed to help explain the smaller change in sea-ice volume in HR, $\Delta E = \Delta E_{LR} - \Delta E_{HR} = (1.79 - 0.49) \times 10^{21} = 1.3 \times 10^{21}$ J, thus allowing to moderate the sea-ice decline in the Southern Ocean.

**Mean state dependence of sea-ice changes**. Bryan et al.[32] argue that the stronger sea-ice volume loss in LR compared to HR (CCSM3.5 model) is not because the LR ice extent decreases more than the HR; instead, they attribute this to a basic property of sea-ice thermodynamics, that is, thicker sea ice decays faster than thinner sea ice[32,81]. However, this thermodynamic argument is only part of the explanation for the different sea-ice behaviour in our simulations with AWI-CM as explained below.

To this end, we choose a mean state (2020–2049, Supplementary Figs. 15 and 16) that is similar in AWI-CM-LR and AWI-CM-HR for both thickness (LR volume is about 23% larger than in HR) and concentration (LR extent: 18.3 million km$^2$, HR extent: 17.84 million km$^2$). Despite the very similar mean states in 2020–2049, the thickness change patterns relative to the 2020–2049 baseline are very different for LR and HR (Supplementary Fig. 17). More than half of the volume changes occur at or in the vicinity of the ice edge (LR and HR, outside of the area of thickest/compact ice as diagnosed from the $0.75 \le$ SIC region, Supplementary Fig. 17). This implies that a large fraction of the volume decrease happens where the largest changes in concentration occur, and not only where the majority of the thick and compact ice is located.

To put the mean state differences for AWI-CM in 2020–2049 into perspective, it is worth noting that the sea-ice volume of the LR and HR models in Bryan et al.[32] was much larger (by as much as 46%) for their LR compared to their HR configuration. When compared also to typical differences between CMIP models, the mean state differences between AWI-CM-LR and AWI-CM-HR in 2020–2049 of 0.46 million km$^2$ are relatively small. For the 2020–2049 period, the CMIP5 multi-model mean (MMM) SIE amounts to $16.1 \pm 4.2$ million km$^2$ (mean ± one standard deviation). For sea-ice volume, the MMM for the 2020–2049 period ranges even more widely: its standard deviation amounts to $8.47 \times 1000$ km$^3$, which is about 58% of the MMM volume ($14.49 \times 1000$ km$^3$). In summary, this shows that a purely thermodynamic argument[32,80,81] is unlikely to be the sole explanation for the different sea-ice behaviour in our simulations.

## Data availability
Antarctic September SIE time series, used in Figs. 1 and 3, are available from NSIDC[97] under (ftp://sidads.colorado.edu/DATASETS/NOAA/G02135/south/monthly/data/). The sea-ice concentration section in Fig. 4a is based on the NSIDC climatology from SMMR and SSM/I-SSMIS, Version 3[98], after registration for an Earthdata Login (https://earthdata.nasa.gov). PHC3.0 climatology data[92] for ocean temperature and salinity can be downloaded from the Polar Science Center (University of Washington) database at http://psc.apl.uw.edu/data/. The raw model data generated during the present study are archived at the German Climate Computing Center (DKRZ) and available from the

corresponding author on reasonable request. The CMIP5 data are openly available from the Earth System Grid Federation (ESGF, https://esgf.llnl.gov/). Historical and RCP8.5 scenario simulation data are also available on the German Climate Computing Centre's (DKRZ) supercomputer Mistral, which is mirrored from the DKRZ node of the ESGF. The data for Fig. 1 are provided in the Source Data file. The data and Jupyter notebooks for Figs. 1–5 in this study are available in the Zenodo database[99] under https://doi.org/10.5281/zenodo.5747692, and in the GitHub repository https://github.com/trackow/AntarcticSeaIce_NatCommun. Source data are provided with this paper.

## Code availability
The source code for the AWI-CM climate model (revision 172) used in this study, including runscripts for the German Climate Computing Centre, is available in the Zenodo database[100] under https://doi.org/10.5281/zenodo.5650135. Jupyter notebooks to reproduce figures are available in the Zenodo database[99] under https://doi.org/10.5281/zenodo.5747692. The archived release v1.0.0 is also publicly available on Github (https://github.com/trackow/AntarcticSeaIce_NatCommun/tree/v1.0.0).

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

## Acknowledgements

This work was supported by the PRIMAVERA project, which has received funding from the European Union's Horizon 2020 research and innovation programme under grant agreement No. 641727 (T.J., D.V.S, T.S.). We also acknowledge funding by the Federal Ministry of Education and Research of Germany in the framework of the research group Seamless Sea Ice Prediction (SSIP), Grant 01LN1701A (H.F.G.). The work was also supported in the framework of the state assignment of the Ministry of Science and Higher Education of Russia, theme No. 0128-2021-0014 (D.V.S.). The work described in this paper has received funding from the Helmholtz Association through the project "Advanced Earth System Model Capacity" in the frame of the initiative 'Zukunftsthemen', ZT-0003 (T.J., T.S.). The figures were done with the graphics environment Matplotlib[95] using Jupyter[96]. We acknowledge the World Climate Research Programme's Working Group on Coupled Modelling, which is responsible for CMIP, and we thank the climate modeling groups for producing and making available their model output. For CMIP, the U.S. Department of Energy's Program for Climate Model Diagnosis and Intercomparison provides coordinating support and led development of software infrastructure in partnership with the Global Organization for Earth System Science Portals. Simulations and analyses were performed at the German Climate Computing Center (DKRZ), in the framework of the project bm0944, on the Mistral supercomputer. We would like to thank Judith Hauck for her feedback and suggestions.

## Author contributions

T.R. wrote the initial manuscript and conceived the study together with T.J. and S.D.. D.V.S. conducted the experiments and T.R. performed the analysis of the results. H.F.G. contributed the CMIP5 analyses and H.H.H. added to the discussion. D.S. coded the routines for the heat transport calculations. All authors contributed to the scientific discussion and reviewed the manuscript.

## Funding

## Competing interests

The authors declare no competing interests.
