## [Peer Review File · Nature Communications]

Delayed Antarctic sea-ice decline in high-resolution climate change simulationsReviewers' Comments:

Reviewer #1 (Remarks to the Author):

Review of Rackow et al., submitted to Nature Comms

The conclusion of this paper is that resolving the ocean mesoscale in the Southern Ocean is critical to reducing the rate of decline in Antarctic sea ice both in the recent period and the future while acknowledging that other factors may contribute. The authors present a compelling argument for why eddy resolving resolution should have this impact on the evolution of Antarctic sea ice by examining the ocean properties and the meridional heat transport in parallel high and low resolution experiments using the AWI coupled model. This work is novel as climate change projections at high resolution have only recently become possible. I think that this paper will enable further work on this topic across the CMIP models to examine the extent that the hypothesis holds. The statistical analysis looks robust and I believe that the work could be reproduced. Overall I would like to see this paper published but I recommend that major revisions are required to address my comments.

Major point 1: impact of mean state

I think that the paper needs greater discussion and investigation of the impact of the differing mean states of Antarctic sea ice in the control simulation and therefore initial conditions of historical plus scenario experiments. While figure 1a doesn't support the mean ice extent being highly correlated with the linear trend, it is noticeable that the mean state in LR has a greater ice extent and that the ice loss starts much earlier. Having said that, extended data figure 9 using the later baseline period of 2020-2049 does show a similar result. To be convinced on this point, I think I would like to see ice extents marked on relevant figures (2, 3c, 4, 5b and d, 9) plus an additional figure showing the mean state of ice extent and thickness. I would also like to see a baseline period chosen where the ice extents and volumes are quite similar – 2020-49 looks a reasonable choice on examination of figure 3a but it would be good to state the relevant extents and volumes for each experiment for that period.

Major point 2: representation of overturning circulation

I appreciate that the authors have used arrows in figure 4 a and b but I think that the authors need to calculate the overturning on density surfaces in the Southern Ocean. This should be calculated using sub-monthly means and will give an improved representation of the circulation.

Minor points:

- Consider changing title to something like 'the impact of resolution on the evolution of Antarctic sea ice' as I think this is more representative to the motivation and discussion which focusses on the recent past as well as the future
- Add results from this study to Figures 1 a and b?

- P2, line 46-p3, line 57 and figure 2. I suggest that this figure should be polar stereographic of the Southern hemisphere and show the ice edges for the two periods, while the discussion would be better focussed on the Southern hemisphere
- P3, line 61-63. Compare also to Hewitt et al, GMD, 2016 which shows increased variability in Weddell Sea sector in high resolution model due to polynyas
- P3, line 64-66. I'm not sure that this is relevant here given the preceding discussion on the spin-up
- Figure 3, add ice edges as discussed above
- P4, line 74. In addition to Weddell Sea, can the authors explain the larger differences in the Ross Sea? Is this responsible for some of the residual variability in HR seen in figure 8?
- P5, line 102. Can the authors expand on the numerical mixing? Is this mixing more significant in LR than HR?
- P7, line 162. Is it possible to put these results into the context of those from larger ensembles? If for example a later baseline was chosen, could the results be explained by internal variability in the climate system?

Helene Hewitt

Reviewer #2 (Remarks to the Author):

Review of "Antarctic sea ice decline delayed well into the 21st century in a high-resolution climate projection" by Rackow et al. in Nature Communications

Summary:

Global climate models typically simulate an almost immediate decline of Antarctic sea ice in response to increasing anthropogenic atmospheric green-house gas concentrations and stratospheric ozone depletion. However, satellite records since about 1980 do not support such an early decline of the Antarctic sea-ice cover in response to the forcing. Rackow et al. here argue that the magnitude and timing of the decline in global climate models is associated with the ocean resolution in these models and that the higher resolution models tend have a delayed and reduced decline of the Antarctic sea-ice cover. The authors present support for this hypothesis from a suit of CMIP5 models as well as contrasting high- and low-resolution experiments with a single climate model. Based on these contrasting experiments with one of the models, they argue that the delay and reduction in the sea-ice cover is associated with eddies that cause differences in the meridional heat transport, which delays the surface warming in the Southern Ocean, and thus the sea-ice decline, more strongly in the high-resolution case. These findings imply that climate models would only be able to accurately simulate the Antarctic sea-ice response to the anthropogenic forcing if they were accurately parameterizing or resolving mesoscale eddies.

Recommendation:

The here presented manuscript presents an interesting and compelling hypothesis that would be of interest to a broad community and would therefore, in principle, be suitable for a publication in Nature Communications. However, as you will see in my detailed comments below, it lacks supporting evidence for many of the claims made in the paper and often contradicts the published literature, which is insufficiently discussed. I feel that the authors are somewhat overselling their results without sufficient and accurate appreciation of the existing literature. A large body of literature has been published on eddy-permitting or eddy-resolving model simulations in the Southern Ocean, even with implications for sea ice, but it is not discussed in relation to the results obtained here. Some of this literature shows that the ocean resolution has no influence on the sea-ice response (Bitz and Polvani, 2012) or that high-latitude Southern Ocean surface warming is intensified due to an increased poleward heat transport in a high-resolution case (Meredith and Hogg, 2006; Fyfe et al., 2007; Screen et al., 2009; Meredith et al., 2015; Griffies et al., 2015), which is opposite to the interpretation provided here. This contradiction, the lack of the discussion on this difference, the missing evidence for the robustness of this finding across the CMIP5 models, makes me worry about the robustness of the results presented here. The resolution of the high-resolution version of the model is not fully eddy resolving, but at the same time the parameterization is switched off. So, it might be that the eddy-heat transport in the high-resolution model is actually even more underestimated (due to the lack of a parametrization and not resolving all eddies) than in the low-resolution case with eddy parametrization and if one would go to even higher resolutions, it would increase again. In addition, the manuscript suffers, in my view, from substantial language and formatting issues (figures), as well as small inaccuracies, which makes it difficult and time-consuming to review and assess. I am hesitant to recommend either major revisions or a rejection with the option to resubmit, given the many issues that I think the authors would need address before one could recommend a publication of this manuscript. Yet, I do think that the overall findings would be highly relevant to the community, if the authors were providing more supporting evidence, the analysis was refined, and the differences to the existing literature were sufficiently addressed and resolved.

Major issues:

1. One of the major issues is that the manuscript is not discussing the previous study by Bitz and Polvani (2012), who is to my knowledge the only other study that investigated the effect of ocean resolution in a climate model on the Antarctic sea-ice response to stratospheric ozone depletion. In contrast to the here presented study, they did not find a substantial effect of the ocean resolution on the Antarctic sea ice response to an external forcing. While I appreciate that the analysis presented here makes use of the entire CMIP5 archive to illustrate the effect of resolution and that Bitz and Polvani (2012) only used one model, I am still wondering why these studies come to such different conclusions and why this difference is not mentioned?

2. Another major issue is the inconsistency with literature and lack of discussion with respect to the Southern Ocean surface warming and meridional heat transport (MHT) response in different resolutions. Numerous studies (Meredith and Hogg, 2006; Fyfe et al., 2007; Screen et al., 2009; Meredith et al., 2015; Griffies et al., 2015) argue that the high-latitude Southern Ocean surface

warming is intensified due to an increased poleward heat transport in a high-resolution case. These findings are opposite to what is presented here.

3. Griffies et al. (2015) and Morrison et al. (2013) also argue that the vertical eddy heat flux is actually a more important aspect resulting from resolving/permitting or not resolving eddies. Morrison et al. (2013) argue that this might be the reason for a cooling tendency in the Southern Ocean. How does the vertical eddy heat flux in these simulations change? How does it compare to the meridional heat flux and how do the results compare to these two studies?

4. Given the points 1-3 above, I am worried that some of these differences to the published literature might result from the fact that the here presented high-resolution (HR) model is not fully eddy resolving, but rather eddy permitting and that at the same time the eddy parametrization is switched off. So, I am concerned that if one would go to even higher resolutions than HR the results would look more like what is being published in literature.

5. Many of the arguments made in the manuscript seem unsupported. For example, in many instances the authors argue for a different response of the residual overturning circulation. However, the residual overturning circulation response is not being shown, even though it could be easily diagnosed from the model. Please note that the residual overturning circulation differs from the meridional heat transport. I think the paper would overall benefit greatly from a more quantitative analysis and statistical significance testing. For example, the differences in the simulations presented in figure 3 and figure 5 do not contain error bounds (e.g. based on natural variability of the control simulations), the differences are not tested for significance, and any of the maps showing changes do not show any significance of the change either.

6. Related to the issue 5, the difference in sea-ice response in the simulation is very assertively being attributed to the MHT (e.g. lines 155 to 158). However, other processes surface or subsurface heat flux changes, dynamical sea ice changes, etc. might also be critical. The methods section (line 225 to 235) seems to attempt to be more quantitative in this attribution. However, it is not extensively discussed in the main text. I would think that if one would want to formally attribute the sea-ice volume response to a mechanism, it would require a full heat budget analysis that should be discussed in the main text.

7. If it was really the difference in MHT causing the different sea-ice response, there should be a similar relation in the CMIP5 models. MHT can easily be obtained from the CMIP5 models. So, I am wondering why the authors do not do a similar analysis as presented in Figure 1 in terms of ocean resolution for the relation between sea-ice change and MHT change? This would certainly provide supporting evidence for their claims if correct.

8. I am also wondering why the CMIP5 results are not more strongly incorporated in the manuscript. The discussion in the main text is very brief and the findings are completely absent from the abstract and conclusions. However, I do think that this is strong evidence for a resolution dependence and the discussion should be expanded. Most certainly this analysis requires a methods section and please follow the guidelines for CMIP5 data usage and citation. It would be also useful to include the AWI HR and LR simulations in Figure 1 for reference.

9. There is an issue with inconsistency to the study by Turner et al. (2013), who report a negative historical September SIE trend for the MRI-CGCM3 model, whereas here it is the one with the most positive trend. Can you explain this different result? Please provide a methods section on how this analysis is being performed.

10. In my view, the manuscript suffers from poor or unclear language and formulation (some examples below). I do not have the time to go into this in detail, but I suggest that the language needs to be improved substantially before I would be able to recommend the publication of this manuscript.

11. Figures: Please use appropriate and consistent font size in all figures, provide all labels and SI units (consistently, e.g. not mixing K and degC) for all figures, and complete x and y axes. Please use a more appropriate color scale for the contour plots (see e.g. <http://www.hclwizard.org/> or <https://doi.org/10.1175/BAMS-D-13-00155.1>). And please follow the journal guidelines.

Minor issues:

Due to time-constraints, I am only listing a couple of minor issues here, but there is a lot more and I feel that the authors should be overall more careful in terms of wording and presentation and follow guidelines more accurately.

- Abstract 2nd sentence (no line numbers provided): Unspecified “This”: What is “this” referring to?
- Abstract: “;” is overused in the abstract and also the text, which makes it much more difficult to read
- Abstract (and later in the text): “a stronger northward branch” is confusing to me. If only the northward branch was stronger but not the southward branch, this would empty out the Southern Ocean in terms of volume. Do you refer to the “upper circulation cell of the [...]”?
- Abstract: Please tone down “a milestone”, which is in my view not a suitable wording for an abstract.
- Throughout the manuscript there is a lot of claim that things are done “for the first time”. In my view this is unnecessary and should be avoided since provides the reader with an uneasy feeling that the authors are overselling their results.
- Line 5: I don’t think reference 5 is most suitable here. The peer-reviewed studies reporting the 2016 sea-ice decline in detail are Turner et al. (2017) and Schlosser et al. (2018).
- Line 6: “some” and “others” needs references
- Lines 8 to 11: I think one of the most widely accepted and not listed hypothesis is that the historical sea-ice expansion is driven by sea-ice dynamics (Holland and Kwok, 2012; Haumann et al., 2014)
- Line 34: What is meant by “not entirely robust in time”?
- Figure 1: How is the grey box defined? Please provide an objective measure, i.e. plus-minus two standard deviations or similar.
- Figure 2: It would be clearer to also show a third panel with the difference between the simulations.
- Lines 49 to 51: The discussion of the North Atlantic and Arctic seem an unnecessary tangent.
- Line 58: Please quantify how different the September sea ice extent is rather than writing “quite different”

- Line 64 to 66: Please reformulate the sentence to provide the reader only with relevant information and do not include subjective measures.
- Line 75 and 80: I do not understand what is meant by “physical nature” and “underlying physics”. What is the alternative if it is not the physics?
- Line 83: I don’t think that one can speak of an “ensemble member” when the models configuration changes. An ensemble member is usually run with the same configuration but slightly perturbed initial conditions.
- Line 84: I would disagree that “the delayed sea-ice decline lies solely in the high-resolution ocean”. It is difficult to say from ED Fig. 10, since the original configuration is not included and no quantitative measure to support this statement is provided. However, the simulations to look different.
- Line 97: It is not “given” that warmer waters will be entrained with time. They could also just occur locally at the subsurface without affecting the surface.
- Line 106: I do not think that “52” is a suitable reference here. Maybe the article by Marshall and Speer (2012) or similar is more suitable.
- Line 109 and 119: I disagree that the upwelling waters are relatively “cold”. They are actually warmer than the surface. If they were cold, there would be no southward heat transport into the upwelling region.
- Lines 110 to 112: The latitudinal ranges of upwelling and subduction are not correct. Please refer to literature for more suitable values.
- Line 110: “nutrient-rich” seems out of context
- Line 112 to 114: These two lines seem out of context and distract the reader from the main story. If anything, they might occur in the introduction or conclusion sections.
- Line 119: I do not see any evidence of realistic formation of deep and bottom waters. Please provide supporting evidence for such a statement.
- Line 122: the statement in parenthesis seems out of context.
- Line 127: If MHT is generally poleward, why should the upwelling water than be relatively “cold” (line 110 and 119)
- Line 129: “northward residual branch”: Note the difference between MHT and residual overturning circulation.
- Line 134: Figure 2 does not show sea-surface temperature but 2-m air temperature. Please add a figure of SST or replace figure 2.
- Lines 144-153: Note the difference between MHT and residual overturning circulation. Please use precise and accurate formulations. Also plot the residual overturning circulation and the differences to provide supporting evidence. “entirely unclear” is not correct, since there is existing literature (see major issues above). Note that eddy saturation is not relevant to this discussion (see e.g. Rintoul, 2018).
- Line 157: I don’t think that it could be claimed that “this difference alone is sufficient”. There would need to be supporting evidence from the entire CMIP5 ensemble.

- Lines 155-189: Please avoid colloquial language.
- Line 186: The effect of “better-resolved bathymetry” is not explained anywhere in this manuscript and I am not sure how this relates here.
- Lines 188-189: I don’t think that such far reaching claims can be made. Please reformulate.
- Lines 236-240: Please revise the Data availability statement to comprise with journal guidelines. I don’t think “on reasonable request” is in line with the guidelines.
- Line 279: I don’t think that this is an appropriate (peer-reviewed) reference here.

References:

Bitz, C. M., and Polvani, L. M. (2012), Antarctic climate response to stratospheric ozone depletion in a fine resolution ocean climate model, *Geophys. Res. Lett.*, 39, L20705, doi:10.1029/2012GL053393.

Fyfe, J.C., O.A. Saenko, K. Zickfeld, M. Eby, and A.J. Weaver, 2007: The Role of Poleward-Intensifying Winds on Southern Ocean Warming. *J. Climate*, 20, 5391–5400, <https://doi.org/10.1175/2007JCLI1764.1>

Griffies, S.M., M. Winton, W.G. Anderson, R. Benson, T.L. Delworth, C.O. Dufour, J.P. Dunne, P. Goddard, A.K. Morrison, A. Rosati, A.T. Wittenberg, J. Yin, and R. Zhang, 2015: Impacts on Ocean Heat from Transient Mesoscale Eddies in a Hierarchy of Climate Models. *J. Climate*, 28, 952–977, <https://doi.org/10.1175/JCLI-D-14-00353.1>

Haumann, F. A., Notz, D., and Schmidt, H. (2014), Anthropogenic influence on recent circulation-driven Antarctic sea ice changes, *Geophys. Res. Lett.*, 41, 8429– 8437, doi:10.1002/2014GL061659.

Holland, P., Kwok, R. Wind-driven trends in Antarctic sea-ice drift. *Nature Geosci* 5, 872–875 (2012). <https://doi.org/10.1038/ngeo1627>

Marshall, J., Speer, K. Closure of the meridional overturning circulation through Southern Ocean upwelling. *Nature Geosci* 5, 171–180 (2012). <https://doi.org/10.1038/ngeo1391>

Meredith, M. P., and A. M.Hogg (2006), Circumpolar response of Southern Ocean eddyactivity to a change in the Southern Annular Mode,*Geophys. Res.Lett.*,33, L16608, doi:10.1029/2006GL026499.

Meredith, M.P., A.C. Naveira Garabato, A.M. Hogg, and R. Farneti, 2012: Sensitivity of the Overturning Circulation in the Southern Ocean to Decadal Changes in Wind Forcing. *J. Climate*, 25, 99–110, <https://doi.org/10.1175/2011JCLI4204.1>

Morrison, A. K., O. A. Saenko, A. McC. Hogg, and P. Spence (2013), The role of vertical eddy flux in Southern Ocean heat uptake, *Geophys. Res. Lett.*, 40, 5445–5450, doi:10.1002/2013GL057706

Rintoul, S.R. The global influence of localized dynamics in the Southern Ocean. *Nature* 558, 209–218 (2018). <https://doi.org/10.1038/s41586-018-0182-3>

Schlosser, E., Haumann, F. A., and Raphael, M. N.: Atmospheric influences on the anomalous 2016 Antarctic sea ice decay, *The Cryosphere*, 12, 1103–1119, <https://doi.org/10.5194/tc-12-1103-2018>, 2018.

Screen, J.A., N.P. Gillett, D.P. Stevens, G.J. Marshall, and H.K. Roscoe, 2009: The Role of Eddies in the Southern Ocean Temperature Response to the Southern Annular Mode. *J. Climate*, 22, 806–818, <https://doi.org/10.1175/2008JCLI2416.1>

Turner, J., T.J. Bracegirdle, T. Phillips, G.J. Marshall, and J.S. Hosking, 2013: An Initial Assessment of Antarctic Sea Ice Extent in the CMIP5 Models. *J. Climate*, 26, 1473–1484, <https://doi.org/10.1175/JCLI-D-12-00068.1>

Turner, J., Phillips, T., Marshall, G. J., Hosking, J. S., Pope, J. O., Bracegirdle, T. J., and Deb, P. (2017), Unprecedented springtime retreat of Antarctic sea ice in 2016, *Geophys. Res. Lett.*, 44, 6868– 6875, doi:10.1002/2017GL073656.

Response to reviewers

We would like to start by expressing our gratitude to the reviewers for their thorough and extremely helpful reviews. Following their comments and recommendations, we have revised the paper substantially over the last year. The most important changes made to the manuscript can be summarized as follows:

1. We considered all the papers suggested by the reviewers in depth. In this process, we also developed a much better understanding of the role of resolution in the representation of Antarctic sea ice changes in a warming world, which should add to the quality of the paper. Furthermore, by accounting for all relevant literature we are now able to put our findings into a better context regarding previously published work. Importantly, as explained below, in our opinion the suggested literature does not contradict our findings, especially in terms of the concerns raised by reviewer 2.
2. Unfortunately, only monthly output from our simulations was stored (as mentioned in the Methods section). Therefore, we cannot compute any diagnostics that require sub-monthly output (such as eddy overturning in density coordinates and eddy heat flux) in a satisfactory way. Therefore, we still refer to salinity sections in the paper as a proxy for the overturning (see Armour et al. 2016 for a rationale of this approach). For future simulations with the successor model version FESOM2, we will be able to carry out the suggested diagnostics online.
3. Both reviewers appreciated the message conveyed by Figure 1, agreeing that it provides strong evidence for a resolution-dependence of Antarctic sea ice changes. Following comment #9 by reviewer #2 about the MRI-CGCM3 model, the underlying CMIP5 trend analysis was refined and completely redone for all ensemble members. Importantly, this revision further increases the statistical significance of the CMIP5 SIE trend dependence on the ocean resolution.
4. We added an entirely new analysis of meridional ocean heat transport (MHT) changes towards the end of the 21st century in CMIP5 models. This turned out to be a non-trivial task due to the many different ocean grids used. As a result, we found that in the coarse-resolution range of the CMIP5 models, the relatively coarser-resolution CMIP5 models show a pronounced southward MHT increase south of 65°S, whereas the relatively higher-resolved CMIP5 models show a weaker increase (between AWI-CM-HR and AWI-CM-LR), consistent with our proposed mechanism.
5. For the important changes shown in Fig.3 and Fig.5, we added significance hatching based on whether the changes are “physically significant” (i.e., larger than the internal variability in the control runs), confirming that the changes and the different response between LR and HR is unlikely to be due to sampling uncertainty related to natural variability.
6. The Discussion section was rewritten to include possible other mechanisms that could explain part of the resolution dependence.

Attached to this letter you find our detailed point-by-point answers to the reviewers' comments. We are confident that you will find our manuscript substantially improved.

Best regards, Thomas Rackow / *On behalf of all co-authors*

Reviewer #1 (Remarks to the Author):

Review of Rackow et al., submitted to Nature Comms

The conclusion of this paper is that resolving the ocean mesoscale in the Southern Ocean is critical to reducing the rate of decline in Antarctic sea ice both in the recent period and the future while acknowledging that other factors may contribute. The authors present a compelling argument for why eddy resolving resolution should have this impact on the evolution of Antarctic sea ice by examining the ocean properties and the meridional heat transport in parallel high and low resolution experiments using the AWI coupled model. This work is novel as climate change projections at high resolution have only recently become possible. I think that this paper will enable further work on this topic across the CMIP models to examine the extent that the hypothesis holds. The statistical analysis looks robust and I believe that the work could be reproduced. Overall I would like to see this paper published but I recommend that major revisions are required to address my comments.

Major point 1: impact of mean state

I think that the paper needs greater discussion and investigation of the impact of the differing mean states of Antarctic sea ice in the control simulation and therefore initial conditions of historical plus scenario experiments. While figure 1a doesn't support the mean ice extent being highly correlated with the linear trend, it is noticeable that the mean state in LR has a greater ice extent and that the ice loss starts much earlier. Having said that, extended data figure 9 using the later baseline period of 2020-2049 does show a similar result. To be convinced on this point, I think I would like to see ice extents marked on relevant figures (2, 3c, 4, 5b and d, 9) plus an additional figure showing the mean state of ice extent and thickness. I would also like to see a baseline period chosen where the ice extents and volumes are quite similar – 2020-49 looks a reasonable choice on examination of figure 3a but it would be good to state the relevant extents and volumes for each experiment for that period.

We thank the reviewer for these helpful comments. We agree that Figure 1 does indeed show that the mean ice extent is not likely to be highly correlated with the linear trend. Nevertheless, we still performed the suggested, more detailed analysis on potential mean state impacts. To this end, we have addressed the 2020-2049 baseline more closely; we quantified the according extents and volumes below; and we have added the suggested additional figure to the supplementary material.

Furthermore, we have added an overlapping 40-year period (2019-2058) for AWI-CM-LR and AWI-CM-HR to Figure 1 to also consider a period with similar extents and volumes. Also for this period, the sea ice extent trends in HR and LR are different by more than 0.4 million square km / decade (see discussion below).

We also added the ice edge contours to Figure 3c (and in ED Figure 9, which is now Supplementary Fig. S6), while for Figure 2 the plot appeared too busy to us. For Figure 4a-c, we also added sections of present-day sea ice fraction from the NSIDC climatology (Stroeve

and Meier 2018) as well as the observed location of the ice edge. We added the same sea ice information for HR and LR.

As suggested by the reviewer, we also added zonal-mean sea-ice sections (and the location of the ice edge) to Figure 5. Since we added a novel CMIP5 analysis of meridional heat transport changes in Figure 5, we also added the CMIP5 multi-model mean location of the ice edge in Fig.5e.

Additional figure (Supplementary Figure S11) showing the LR mean states of ice extent and thickness. 30yr-mean sea ice concentration and thickness for (left) 2020-2049

and for (right) 2070-2099 in the LR simulation. White solid contour is always sea ice extent (SIC>15%) in 2020-2049. The dashed contours indicate sea ice extent for 2070-2099.

Additional figure (Supplementary Figure S12) showing the HR mean states of ice extent and thickness. 30yr-mean sea ice concentration and thickness for (left) 2020-2049 and for (right) 2070-2099 in the HR simulation. White contour is always sea ice extent (SIC>15%) in 2020-2049. The dashed contours indicate sea ice extent for 2070-2099.

For the 2020-2049 period, the mean states are indeed similar, with **extent**=18.3 million square km for LR and 17.84 million square km for HR. The **volume** in LR is about 23% larger than in HR, but not as large as the 46% mentioned in Bryan et al (2014).

These differences are also smaller than what is typically observed in the CMIP5 models. For 2020-2049, the CMIP5 multi-model mean (MMM) extent is 16.1 +/- 4.2 million square km. For volume, the MMM for 2020-2049 ranges even more widely: the standard deviation (8.47 x 1000 km³) amounts to 58% of the MMM volume (14.49 x 1000 km³). This does, however, not lead to significantly delayed sea ice decline in the CMIP5 models.

To further explore the hypothesis that the sea ice mean state dependence is not the leading factor, we checked in more detail where the sea ice volume is lost. Is it due to the decreasing extent, or is it mostly decreasing where the thick ice is? This is answered in the following paragraphs:

- The volume reduction in LR is from 11.35 * 1000 km³ (2020-2049) to 7.61 * 1000 km³ (2070-2099), so **-3.74 * (1000 km³)**. The extent decrease accounts for 19.5% of the volume decrease.
- The volume reduction in HR is from 9.21 * 1000 km³ (2020-2049) to 7.29 * 1000 km³ (2070-2099), so only about half of LR with **-1.92 * 1000 km³**. The reduction in sea ice extent accounts for 18.7% of the volume decrease.

However, when extending the area along the sea ice edge, by including the medium-compact ice, we see higher percentages of volume are lost within this extended region. Inside the 0.75 concentration contour, the concentration does not change significantly, and outside most of the changes in concentration are happening in an annular ring. The volume decrease outside of the 0.75 concentration contour (taken for 2070-2099, Supplemental Figure S13) can explain 50.4% of the total volume decrease in LR. The volume decrease in HR northward of this area can explain 54.3% of the total volume decrease.

This implies that a large fraction of the volume decrease happens where the largest changes in concentration occur, and not only where the majority of the thick/compact ice is located (see Bitz et al. 2004).

Overall, we conclude that although thicker ice decays faster than thinner ice, as explained in Bitz et al. (2004), this is only part of the story in our case. In our study, more than half of the volume changes occur at or in the vicinity of the ice edge (in other words outside of the area of thick/compact ice). Moreover, although the mean state is similar in LR and HR in 2020-2049 for both thickness and concentration, the thickness change patterns relative to that period are very different for LR and HR. In our opinion this tells us again that the thermodynamic argument by Bitz et al. (2004) is likely not the leading explanation.

We have added a new section to the Methods ("*Mean-state dependence of sea-ice changes*", l. 271-310) where details of above analysis are given.

References:

Bitz, C. M. & Roe, G. H. A mechanism for the high rate of sea ice thinning in the Arctic Ocean. *J. Clim.* 17, 3623–3632 (2004)
[https://doi.org/10.1175/1520-0442\(2004\)017<3623:AMFTHR>2.0.CO;2](https://doi.org/10.1175/1520-0442(2004)017<3623:AMFTHR>2.0.CO;2)

Major point 2: representation of overturning circulation

I appreciate that the authors have used arrows in figure 4 a and b but I think that the authors need to calculate the overturning on density surfaces in the Southern Ocean. This should be calculated using sub-monthly means and will give an improved representation of the circulation.

Only monthly output data was stored for our high-resolution experiments (we mention this in the Methods). This makes it challenging to compute the suggested diagnostics, which require sub-monthly output to be really convincing.

Although also mentioned by reviewer 2, giving the overturning in density coordinates is in any case not straightforward for version 1 of FESOM (which is the ocean-sea ice model in the AWI-CM configuration used here). There are some difficulties with diagnosing transports/streamfunctions in isopycnal coordinates residing in the finite-element discretization of the model. This is worked on in our group and addressed in the successor model FESOM2, as discussed in more detail in Sidorenko et al. (2020). Since the overturning on z-levels is known to be dominated by the “Deacon cell” in the relevant latitude range, we decided not to include it in the revised paper.

Therefore, we still refer to salinity sections in the paper as a proxy for the overturning circulation, as is sometimes done in the literature (e.g. in Armour et al. 2016; their Fig. 1), and give schematic arrows for the implied streamfunction. We will be able to diagnose the suggested diagnostics in more detail in future simulations with the newer model FESOM2 that employs a finite volume approach.

References:

Sidorenko, D., Danilov, S., Koldunov, N., Scholz, P., and Wang, Q.: Simple algorithms to compute meridional overturning and barotropic streamfunctions on unstructured meshes, Geosci. Model Dev., 13, 3337–3345, <https://doi.org/10.5194/gmd-13-3337-2020>, 2020.

Armour, K. C., Marshall, J., Scott, J. R., Donohoe, A. & Newsom, E. R. Southern Ocean warming delayed by circumpolar upwelling and equatorward transport. Nat. Geosci. 9, 549–555 (2016). DOI 10.1038/ngeo2731. <https://doi.org/10.1038/ngeo2731>

Minor points:

- Consider changing title to something like ‘the impact of resolution on the evolution of Antarctic sea ice’ as I think this is more representative to the motivation and discussion which focusses on the recent past as well as the future

Thank you for this suggestion. We would like to leave this decision to the editor since the term “evolution” also appears slightly vague to us, but we tentatively changed the title to “Delayed Antarctic sea ice decline in high-resolution climate change simulations”. By avoiding the term „projections“, it includes the analysis of the recent past.

- Add results from this study to Figures 1 a and b?

Thank you, this is a good idea and we have done so! Please see more detailed explanations below.

First of all, following comment #9 by reviewer #2 about the MRI-CGCM3 model, the underlying CMIP5 trend analysis in Fig.1 is now carried out for all ensemble members. Given the larger sample size, this led to further increased statistical significance of the CMIP5 SIE trend dependence on ocean resolution ($p=0.034 \rightarrow p=0.011$).

Following the reviewer’s suggestions, we added results of AWI-CM-LR and HR (about 63km and 14km average resolution in the 45–65°S latitude band, respectively) to Figure 1. For the LR and HR models, we decided to be careful by only adding later 40-year trends for the period 2019-2058, for which LR and HR show a similar mean state, and for a time that is arguably not affected by the spin-up anymore (other periods are shown in Supplemental Figure S2 and S3). Even for this much later time period, HR stays in the group of “blue” models (trend close to OBS), while LR joins the “red” models (with strong negative trends) from the beginning. Trends in the historical period 1979-2018 in AWI-CM are discussed below.

If we compute a “running-trend” for our simulations, it turns out that during 1979-2018, HR is (by chance due to multi-decadal variability associated with the spin-up) in a low sea ice extent state, similar to that for LR:

This means that LR and HR trends are very similar for 1979-2018, because the (spin-up-related) peaks of variability in HR occur in a way that the 40-year trend is negative. A clear difference between LR and HR is that the latter recovers from this low-extent state, and that the actual start of strong externally forced negative trend is delayed to the middle of the 21st century (when HR reaches a rate of decrease similar to what LR showed throughout most of the simulation, i.e. -0.5 million km² per decade or stronger).

- *P2, line 46-p3, line 57 and figure 2. I suggest that this figure should be polar stereographic of the Southern hemisphere and show the ice edges for the two periods, while the discussion would be better focussed on the Southern hemisphere*

We added difference panels to Figure 2 (panels c and f) in polar stereographic projection, as also suggested by reviewer 2, which show the differences in the Southern Hemisphere clearly.

We kept the other global plots as they illustrate the fact that global climate change projections are considered. Furthermore, global plots clearly highlight that the impact of ocean resolution is largest in the Southern Ocean, at least for the configurations considered here. The remainder of the paper focuses exclusively on the Southern Ocean.

- *P3, line 61-63. Compare also to Hewitt et al, GMD, 2016 which shows increased variability in Weddell Sea sector in high resolution model due to polynyas*

We mention this paper as follows (l. 93-95): “*The high-resolution experiment HR, on the other hand, exhibits enhanced decadal-scale variability in sea-ice extent between 1950–1990 (especially in the Weddell Sea sector; Supplementary Fig. S4), which has been found in other high-resolution models (Hewitt et al., 2016).*”

- *P3, line 64-66. I’m not sure that this is relevant here given the preceding discussion on the spin-up*

We removed the comparison to the simulation and just mention what is seen in reconstructions. We phrased this as follows (l. 97-100): “*Interestingly, there is some evidence for multi-decadal sea ice variability from reconstructions (53–55). These suggest a 20th century decline of Antarctic sea ice extent, by up to 20% between the 1950s and 1990s (54), which precedes the later increase found in post-1979 satellite data (55).*”

- *Figure 3, add ice edges as discussed above*

We added the sea ice extents to Figure 3 as suggested by the reviewer.

- *P4, line 74. In addition to Weddell Sea, can the authors explain the larger differences in the Ross Sea? Is this responsible for some of the residual variability in HR seen in figure 8?*

Yes, sea ice concentration in the Ross Sea is more variable in HR than in LR, which can also be seen in the internal variability estimate we performed (see Methods and Supplementary Figure S10) based on the LR (panel a) and HR (panel b) control simulations, and this can explain part of the residual variability:

- *P5, line 102. Can the authors expand on the numerical mixing? Is this mixing more significant in LR than HR?*

Analyses of watermass transformations in the North Atlantic in simulations performed on the LR and HR grids (Sidorenko et al., 2020, <https://agupubs.onlinelibrary.wiley.com/doi/10.1029/2020MS002317>) show that spurious mixing is higher on the coarser mesh. We can not repeat such analysis for the simulations reported here because the fields needed for it, as well as for diagnosing temperature tendencies due to isoneutral mixing, have not been stored.

Spurious mixing occurs due to dissipative truncation errors in the advection operators. Such errors are smaller on finer meshes, which suggests that spurious mixing will be also smaller. A caveat here is that an increased eddy variability on finer meshes may counteract this decrease in truncation errors at the grid scale, making the final answer uncertain.

We adjusted the text in the manuscript to stress that spurious mixing can be an unaccounted factor (l. 135-142): *“The upwelled water of Atlantic origin ($\sigma_2 \approx 37.1$) extends too far south in the low-resolution experiment, probably in part due to numerical diapycnal mixing, leading to the relatively warm and salty subsurface conditions. Another contribution could be the joint functioning of the eddy parameterization, which acts to flatten isopycnals, and isoneutral (Redi) diffusivity. This latter mixing along isopycnals is not well constrained and might lead to diffusive transport of warmer waters. A similar statement as for the upwelled waters holds for lighter waters ($\sigma_2 \approx 36.1$ – 36.4) that reach under the sea ice in LR (white contours in Fig.4c), which is in contrast to the observed outcropping northwards of the sea-ice edge in the observations and in HR (Fig.4a,b).*

Again, HR is able to maintain a sharper density contrast between the water masses of Atlantic and Antarctic origin in the 1000–2000 m range.”

- *P7, line 162. Is it possible to put these results into the context of those from larger ensembles? If for example a later baseline was chosen, could the results be explained by internal variability in the climate system?*

This paragraph is not part of the rewritten Discussion section anymore. However, several studies have shown that the discrepancy between modelled and observed Antarctic sea ice trends most likely cannot be explained by internal variability (refer e.g. to the study by Chemke & Polvani (2020), where large ensembles are studied). Instead, the authors conclude that systematic errors are responsible:

“We show that internal variability cannot account for the discrepancy, which therefore is likely to stem from biases in the models' forced response to the external forcing.”

This is consistent with the papers by Jones (2016) and Rosenblum & Eisenman (2017), in which the importance of systematic errors is hypothesized as well.

In addition, as a response to the reviewer's comment we now compare the simulated sea ice and potential temperature changes under global warming in Fig.3 and Fig.5 to estimates of internal variability in the 1950-control simulations (Supplemental Fig. S10). The change of the response (Fig.5f) is significant and cannot be explained by natural variability (see hatching for the regions where natural variability could explain the difference in the response).

References:

- Chemke, R., & Polvani, L. M. (2020). Using multiple large ensembles to elucidate the discrepancy between the 1979–2019 modeled and observed Antarctic sea ice trends. *Geophysical Research Letters*, 47, e2020GL088339. <https://doi.org/10.1029/2020GL088339>
- Jones, J. M. et al. Assessing recent trends in high-latitude Southern Hemisphere surface climate. *Nat. Clim. Chang.* 6, 917–926 (2016). DOI 10.1038/nclimate3103. <https://doi.org/10.1038/nclimate3103>.
- Rosenblum, E., & Eisenman, I. (2017). Sea Ice Trends in Climate Models Only Accurate in Runs with Biased Global Warming, *Journal of Climate*, 30(16), 6265-6278. <https://doi.org/10.1175/JCLI-D-16-0455.1>

Reviewer #2 (Remarks to the Author):

Review of “Antarctic sea ice decline delayed well into the 21st century in a high-resolution climate projection” by Rackow et al. in Nature Communications

Summary:

Global climate models typically simulate an almost immediate decline of Antarctic sea ice in response to increasing anthropogenic atmospheric green-house gas concentrations and stratospheric ozone depletion. However, satellite records since about 1980 do not support such an early decline of the Antarctic sea-ice cover in response to the forcing. Rackow et al. here argue that the magnitude and timing of the decline in global climate models is associated with the ocean resolution in these models and that the higher resolution models tend have a delayed and reduced decline of the Antarctic sea-ice cover. The authors present support for this hypothesis from a suit of CMIP5 models as well as contrasting high- and low-resolution experiments with a single climate model. Based on these contrasting experiments with one of the models, they argue that the delay and reduction in the sea-ice cover is associated with eddies that cause differences in the meridional heat transport, which delays the surface warming in the Southern Ocean, and thus the sea-ice decline, more strongly in the high-resolution case. These findings imply that climate models would only be able to accurately simulate the Antarctic sea-ice response to the anthropogenic forcing if they were accurately parameterizing or resolving mesoscale eddies.

Recommendation:

The here presented manuscript presents an interesting and compelling hypothesis that would be of interest to a broad community and would therefore, in principal, be suitable for a publication in Nature Communications. However, as you will see in my detailed comments below, it lacks supporting evidence for many of the claims made in the paper and often contradicts the published literature, which is insufficiently discussed. I feel that the authors are somewhat overselling their results without sufficient and accurate appreciation of the existing literature. A large body of literature has been published on eddy-permitting or eddy-resolving model simulations in the Southern Ocean, even with implications for sea ice, but it is not discussed in relation to the results obtained here. Some of this literature shows that the ocean resolution has no influence on the sea-ice response (Bitz and Polvani, 2012) or that high-latitude Southern Ocean surface warming is intensified due to an increased poleward heat transport in a high-resolution case (Meredith and Hogg, 2006; Fyfe et al., 2007; Screen et al., 2009; Meredith et al., 2015; Griffies et al., 2015), which is opposite to the interpretation provided here. This contradiction, the lack of the discussion on this difference, the missing evidence for the robustness of this finding across the CMIP5 models, makes me worry about the robustness of the results presented here. The resolution of the high-resolution version of the model is not fully eddy resolving, but at the same time the parameterization is switched off. So, it might be that the eddy-heat transport in the high-resolution model is actually even more underestimated (due to the lack of a parametrization and not resolving all eddies) than in the low-resolution case with eddy parametrization and if one would go to even higher resolutions, it would increase again.

In addition, the manuscript suffers, in my view, from substantial language and formatting issues (figures), as well as small inaccuracies, which makes it difficult and time-consuming to review and assess. I am hesitant to recommend either major revisions or a rejection with the option to resubmit, given the many issues that I think the authors would need address before one could recommend a publication of this manuscript. Yet, I do think that the overall findings would be highly relevant to the community, if the authors were providing more

supporting evidence, the analysis was refined, and the differences to the existing literature were sufficiently addressed and resolved.

We thank the reviewer for the assessment of our paper and the many suggestions that clearly improved our manuscript. Below, we will provide more supporting evidence through an extended CMIP5 analysis. We also refined the other analyses over the last year (for example by checking for physical significance of presented changes), and demonstrate that the suggested additional literature actually often fits rather well to our findings. Additionally, we discuss some of the listed papers more closely that – at first sight – seem to contradict our findings.

We have rewritten the Discussion where we have toned down the attribution and openly discuss other potential contributors besides meridional heat transport.

Major issues:

1. *One of the major issues is that the manuscript is not discussing the previous study by Bitz and Polvani (2012), who is to my knowledge the only other study that investigated the effect of ocean resolution in a climate model on the Antarctic sea-ice response to stratospheric ozone depletion. In contrast to the here presented study, they did not find a substantial effect of the ocean resolution on the Antarctic sea ice response to an external forcing. While I appreciate that the analysis presented here makes use of the entire CMIP5 archive to illustrate the effect of resolution and that Bitz and Polvani (2012) only used one model, I am still wondering why these studies come to such different conclusions and why this difference is not mentioned?*

We thank the reviewer for mentioning the Bitz and Polvani (2012) paper. Despite the different external forcing and considered timescale (about 50 years in order to study the quasi-equilibrated impact of ozone depletion in the 20th century, which is different to our focus of a long-term projection until 2099), the paper is clearly of interest. We had been citing the later Bryan et al. (2014) study, who apply the same 0.1° vs 1° CCSM3.5 configurations with 1% per year increasing carbon dioxide concentrations, but we overlooked the previous Bitz and Polvani (2012) discussion. The paper is now mentioned in our manuscript and their findings fully integrated as detailed below.

We also want to thank the reviewer for the positive assessment that the entire CMIP5 archive enters into our motivational Figure 1. Following another suggestion, this fact is now even strengthened by using all available members (which lends further weight to our “observation” of the role of model resolution). Using all ensemble members also helped to explain the “inconsistency” to the published paper by Turner et al. (see reviewer #2’s comment below).

In the following, we provide a summary of the Bitz and Polvani (2012) paper and put their results in context of our study.

The authors perform a specific warming experiment, an “Ozone Depletion” experiment (OD), at ocean resolutions of 0.1° (also named “HR”) and 1° (“LR”). The resulting warming is accompanied with an increase in westerly winds in the Southern Ocean due to the OD. Their

HR configuration warms less than their LR (along and northward of the sea ice edge), which is in fact a similar finding as in our AWI-CM-HR and LR simulations.

However, differences in the magnitude of the (annual-mean) warming response due to OD in LR and HR (mostly only up to $0.5-1^{\circ}$) are very different to what we are seeing when running a full high-emission climate change scenario (combining depletion of ozone + increased radiative forcing). Compared to LR, our high-resolution temperature response is generally lower by 1–2 K, and locally reduced by up to 4K in the DJF (December-January-February) season, and by about 5K in JJA.

Furthermore, Bitz and Polvani (2012) report subsurface changes that are different from our simulations:

- The HR configuration of Bitz and Polvani (2012) warms less than their LR at the surface. This is a similar finding as in our AWI-CM-HR and LR simulations, although the magnitude of the response is very different.
- If not for an additional warming directly under the ice in their HR (their Fig.3), the ice decline would thus likely be less pronounced than in their LR simulation.
- We do not have a similar warming directly under the ice in HR (it stays in the sub-surface, our Figure 5c and 5f), which can in principle explain the difference between our study and Bitz and Polvani (2012).
- It is not obvious to us why this warming directly under the ice should be more pronounced in their HR simulation than in their LR simulation.
- The reaction of MHT at around 50°S shows that northward transport increases stronger in their LR than HR. This leads us to assume that the GM is tuned differently from our case, and there is a stronger reaction to winds (i.e. GM is weak).
- Bryan et al. (2014) also note that the sea ice loss percentages shown in Bitz and Polvani's Fig. 1 are considerably smaller, only about 25%–30% of those in Bryan et al.'s Fig. 5 and Table 1 (with $2\times\text{CO}_2$ applied for both their LR and HR). It is important to note that Bryan et al. (2014) appear to agree on the different sign of MHT response in LR and HR configurations.

In summary, the differences between Bitz and Polvani (2012) and our results can partly be explained by differences in the applied model forcing and in the scientific questions that have been addressed. Remaining differences are, in our opinion, another manifestation of the fact that in lower-resolution runs we rely on parameterizations (e.g. for the effect of eddies with GM + Redi) and may observe a different behavior depending on the tuning. One is less prone to errors in HR runs; however, even the HR results might depend on yet another parameterization, namely the mixing parameterizations under the sea ice. The different behaviour of the subsurface under the ice in their HR (Figure 3) is a likely candidate for the different ice response in their and our HR simulation. Since the warming signals in Bitz and Polvani (2012) appear to be much smaller than in our study, they might also be less relevant to the sea ice.

The paper by Bitz and Polvani (2012) also pointed us to thermodynamic aspects as discussed by Bitz and Roe (2004), and we have addressed these in the response to

reviewer #1's Major Comment 1 above. Thicker ice decays faster on warming than thinner ice because of more efficient refreezing in winter in the latter case. While this is a valid observation, a valid question is why the different initial sea ice states appear.

We discuss this as follows in the Discussion (l. 240-243): *“An open question is why the different mean states arise in the first place. Since the HR sea-ice extent is closer to the observed extent than LR without particular tuning, it is possible that the different sea-ice mean state in LR itself is affected by details of the eddy parameterization and the choice of GM coefficient, which has been shown to determine the pattern of warming south of the ACC (69).”*

2. Another major issue is the inconsistency with literature and lack of discussion with respect to the Southern Ocean surface warming and meridional heat transport (MHT) response in different resolutions. Numerous studies (Meredith and Hogg, 2006; Fyfe et al., 2007; Screen et al., 2009; Meredith et al., 2015; Griffies et al., 2015) argue that the high-latitude Southern Ocean surface warming is intensified due to an increased poleward heat transport in a high-resolution case. These findings are opposite to what is presented here.

We thank the reviewer for pointing out these studies, which we will discuss in the context of our findings further below. Please note that differences can arise due to model tuning and settings, or experimental setup/scientific focus, which can make it difficult to directly compare with our study. We have now considered these studies and argue below that they are not opposite to what we are presenting.

It is true that some previous studies have reported increased poleward heat transport in high-resolution simulations compared to low-resolution simulations, e.g. in uncoupled ocean simulations (Screen et al. 2009) or in simplified quasi-geostrophic ocean model simulations (Meredith and Hogg, 2006). In these cases, a warmer Southern Ocean mean state can result in high-resolution compared to low-resolution simulations (Meredith and Hogg, 2006). However, this finding depends on how the low-resolution ocean is tuned. In a climate model of intermediate complexity (Fyfe et al., 2007), it was shown that the pattern of warming depends on GM, with increasing GM coefficients leading to an increase of warming south of the ACC. In principle, a high-resolution model can thus be cooler in the Southern Ocean than the low-res version when the parameterized eddy-induced transport is strong. This appears to be the case in AWI-CM-LR with its nominally 1° ocean grid. In fact, the GM coefficient in our low-res model is highly variable and on the high side with up to approx. $2000 \text{ m}^2 \text{ s}^{-1}$ for a 100 km wide grid cell, with vertical scaling proportional to buoyancy frequency. The GM coefficient for the LR grid in AWI-CM was chosen in order to minimize biases with respect to zonally-averaged Southern Ocean temperatures in ocean-only simulations (Fig.6 in Downes et al., 2015), and it lies within the typical range of ocean models (Table 1 in Farneti et al., 2015). For the HR grid in AWI-CM, the coefficient is zero everywhere in the Southern Ocean, i.e., eddies are explicitly simulated, and the Southern Ocean south of the ACC is cooler than in LR (Fig. 4).

While we show that the poleward heat transport and the mean states in the control simulations can already be resolution-dependent, the response of eddies and the associated heat transport in a warming climate is the focus of our study.

Externally-forced greenhouse gas (GHG) increase model experiments as well as ozone depletion experiments are both typically associated with increased Southern Hemisphere westerly winds. Under the influence of increased westerly winds, an increase in eddy kinetic energy will imply a stronger eddy-induced circulation, which is southward in the upper part of the ocean (see papers listed by the reviewer). The total heat transport change is, however, determined by the extent to which explicitly-resolved or parameterized eddies oppose the increasing Ekman transport towards the north. This can be either through partially opposing (net northward transport change), fully compensating (near-neutral transport change), or by reacting with a possible overcompensation (net southward transport change).

Somewhat unexpectedly, as mentioned also by the reviewer, a high-resolution simulation with explicit simulation of meso-scale eddies can show less southward heat transport (i.e., a neutral or even northward heat transport change) compared to its control in response to increasing GHG and associated enhanced winds, as demonstrated by the different sign of MHT change in our dedicated HR and LR projections. This behaviour was also found in previous high-resolution ozone depletion (Bitz and Polvani, 2012) and CO₂ doubling experiments (Bryan et al. 2014):

- Bryan et al. (2014) and Bitz and Polvani (2012) actually agree with the results of our study on the different sign of the ocean MHT change due to different resolutions in warming experiments. The sea ice response is delayed accordingly in our AWI-CM-HR simulation when compared to LR. Potentially due to the different subsurface conditions in Bitz and Polvani (2012, their Fig.3), their HR sea ice declines, however, similarly to their LR, as discussed above; in the light of the many differences between LR and HR, Bitz and Polvani (2012) themselves describe their result as “surprising”: *“In spite of these differences, and although the area of significant surface warming is larger in the 1° case, we obtain the surprising results that the total annual loss of Antarctic SIE is nearly independent of resolution, at $0.77 \times 10^6 \text{ km}^2$ in the 0.1° case and $0.85 \times 10^6 \text{ km}^2$ in the 1° case.”*

- Bitz and Polvani (2012) suspect that *“the regions of strong eddy compensation are too far north to be of much consequence to the sea ice”*. Their maximum southward total MHT anomaly is at about 60°S in LR, largely due to eddies (Fig.4d); their HR shows near-zero MHT anomaly at this latitude.

Very similarly, our maximum southward MHT anomaly in LR is about -0.025 PW. It is however positioned more poleward at about 65°S, well within the sea ice-covered area. Our HR ocean shows near-zero ocean MHT change at this latitude, which is not unlike the Bitz and Polvani (2012) result.

- In summary, the more poleward position of the maximum southward MHT anomaly in our LR simulation, as well as the cooler subsurface in our HR, both support a stronger LR decline and a delayed HR sea ice decline. These small but important differences to Bitz and Polvani (2012) suggest once more that details of applied parameterizations (e.g. GM + Redi in LR) are critical for the simulated sea ice

response. In the HR runs without eddy parameterization, details of the mixing parameterization under sea ice are a likely candidate for remaining differences.

We explain this point as follows in the revised paper (l. 180-183): *“In a previous modelling study by Bitz and Polvani (2012) with the CCSM3.5 model, a resolution-dependent additional warming in their 1° case was found mostly northward of about 60°S, away from the ice edge. This finding has been used by the authors as a possible explanation for the more similar sea-ice response (for different ocean resolutions) than might have been anticipated. In contrast to HR, our low-resolution climate change experiment shows increased poleward MHT even south of 60S (Fig. 5d).”*

Furthermore, Screen et al. (2009) use a coarse 1° ocean model with GM coefficient of only $600 \text{ m}^2 \text{ s}^{-1}$ (i.e. not a high value for a $\approx 100 \text{ km}$ wide grid cell), and they study the response to increased westerly winds associated with the Southern Annular Mode (SAM). Consistent with the argument given above that the parameterized eddy-induced transport could thus be relatively low, the high-resolution $1/12^\circ$ ocean setup in Screen et al. (2009) shows a stronger warming southward of the ACC compared to the low-resolution configuration.

As noted by Bryan et al. (2014), zonal wind stress increase experiments, such as the ones by Screen et al. (2009) and in some of the papers listed by the reviewer, are different in that they typically increase the wind stress abruptly by at least 20% or more, while the warming effect of increased GHG concentrations is neglected. In contrast, in the CO_2 doubling experiments by Bryan et al. (2014) and in the RCP8.5 scenarios performed with AWI-CM, wind stress only changes by about 5-7 % over a period of more than hundred years (Supplementary Fig. S8). Passive advection of the warming signal by the residual ocean circulation can delay warming in the Southern Ocean in CMIP5 models both over the satellite era and over a 100-year timescale after CO_2 quadrupling (e.g. Fig. 2 in Armour et al. 2016). As shown by our experiments, the sea ice response during the satellite era and in projections until the end of the 21st century appears to be linked to the associated heat transport, which depends on details of the underlying (residual-mean) ocean circulation.

Additional references:

Downes, S. M. et al. An assessment of Southern Ocean water masses and sea ice during 1988–2007 in a suite of interannual CORE-II simulations. *Ocean. Model.* 94, 67–94 (2015). <https://doi.org/10.1016/j.ocemod.2015.07.022>.

Farneti, R. et al. An assessment of Antarctic Circumpolar Current and Southern Ocean meridional overturning circulation during 1958–2007 in a suite of interannual CORE-II simulations. *Ocean. Model.* 93, 84–120 (2015). <https://doi.org/10.1016/j.ocemod.2015.07.009>.

In the paper we summarize the discussion as follows, where we tried to group the above mentioned papers by common topic and with respect to scientific question (l. 203-211):

“The impact of ocean resolution has been discussed in previous modelling studies (Screen et al. 2009, Meredith and Hogg 2006, Meredith et al. 2012, Fyfe et al. 2007, Bitz and Polvani 2012, Bryan et al. 2014, Griffies et al. 2015, Sein et al. 2018, Rackow et al. 2019) and some of the results, in particular the sign of the MHT response, may seem contradictory to what is concluded here. However, as detailed below, a meaningful comparison with previous studies is surprisingly difficult to carry out given major differences in model forcing (e.g., abrupt change in wind forcing (Screen et al. 2009, Meredith and Hogg 2006) vs. slow change in temperature and winds due to increasing greenhouse gas concentrations (Meredith et al. 2012, Bryan et al. 2014)), model tuning (e.g., strength of the eddy parameterization in low-resolution setups (Fyfe et al. 2007, Rackow et al. 2019)), scientific question (e.g., the isolated impact of ozone depletion (Bitz and Polvani 2012) or the impact of resolution on the mean state (Griffies et al. 2015, Sein et al. 2018, Rackow et al. 2019) vs. response to forcing (Bryan et al. 2014)), and possibly various mechanisms at play (Fyfe et al. 2007, Bryan et al. 2014). Furthermore, what is actually known about the impact of resolution is obtained from a rather limited number of coupled models (namely, e.g., CCSM (Bitz and Polvani 2012, Bryan et al. 2014)), reflecting the fact that coupled climate modelling with eddy-resolving ocean components is still in its infancy.”

With regards to the Southern Ocean surface warming, mentioned by the reviewer, for example Meredith and Hogg (2006) only speculate on the role of poleward (eddy) heat flux in explaining the (mid-depth) warming of the Southern Ocean, as documented in Gille (2002). Gille (2002) mainly discusses changes in the 700m to 1100m range for the period 1990-2000 compared to previous decades. At the surface, a more recent study by Armour et al. (2016) has shown that for 1982-2012 the surface Southern Ocean south of the ACC has in fact generally cooled – and northward of it, it has warmed (Fig.1 in Armour et al. 2016, <https://www.nature.com/articles/ngeo2731#MOESM243>).

3. *Griffies et al. (2015) and Morrison et al. (2013) also argue that the vertical eddy heat flux is actually a more important aspect resulting from resolving/permitting or not resolving eddies. Morrison et al. (2013) argue that this might be the reason for a cooling tendency in the Southern Ocean. How does the vertical eddy heat flux in these simulations change? How does it compare to the meridional heat flux and how do the results compare to these two studies?*

As already mentioned above in a response to reviewer #1, vertical eddy heat fluxes cannot be satisfactorily diagnosed since we did not store sub-monthly output. However, in the Discussion of the revised version we do mention vertical fluxes as another factor that can in principle contribute (l. 236/237):

“More generally, there are other possible mechanisms that could explain part of the resolution dependence, including those that relate to vertical eddy heat fluxes (Morrison et al. 2013, Griffies et al. 2015) or to the influence of differing mean states.”

Referring to Bitz and Polvani (2012), please note that their total vertical heat flux anomaly is almost the same in their LR and HR simulations (their Fig. 4c), and just the partitioning between the mean and eddy response is different. Similarly for Griffies et al. (2015), vertical eddy heat fluxes assume a more prominent role, but the net effect is not obvious since also the mean flux changes. However, because of the increasing role of eddies, Griffies et al. (2015) certainly argue that the correct physics lies in resolving eddies rather than parameterizing their effects.

In line with our study, the more recent study by Armour et al. (2016) suggests *“that meridional ocean heat transport (OHT) changes rather than vertical heat redistribution or SHFs have predominantly shaped the pattern of SO warming.”*

4. *Given the points 1-3 above, I am worried that some of these differences to the published literature might result from the fact that the here presented high-resolution (HR) model is not fully eddy resolving, but rather eddy permitting and that at the same time the eddy parametrization is switched off. So, I am concerned that if one would go to even higher resolutions than HR the results would look more like what is being published in literature.*

We believe that we have clarified the points 1-3 as much as possible from the simulations we performed and the output we stored. We have confidence in our results because even for the 0.1° global resolution in Bitz and Polvani (2012), a similar behaviour of the MHT response mechanism is shown. The subsurface conditions are however different, as mentioned above, which might explain part of the differences between our studies.

Even with a coarse resolution configuration one might hope to be able to tune the Gent McWilliams (GM)-Redi part to get an appropriate behavior. On the other hand, even in fully eddy-resolving models the structure of isopycnals in the Southern Ocean is not necessarily perfect. At high resolution, following Bryan et al. (2014), standing eddies will also become more important, which is in any case fully beyond the scope of the GM parameterization. We highlight this point in the discussion (l. 232-235):

“In their studies with the same eddy-resolving coupled model CCSM3.5, Bryan et al. (2014) also highlight a role for standing eddies, i.e. meanders of the ACC, in balancing the wind-driven overturning. This mechanism of ACC equilibration will need further investigation in future studies, as its effects are fully beyond the scope of what eddy parameterizations in current climate models can achieve.”

5. *Many of the arguments made in the manuscript seem unsupported. For example, in many instances the authors argue for a different response of the residual overturning circulation. However, the residual overturning circulation response is not being shown, even though it could be easily diagnosed from the model. Please note that the residual overturning circulation differs from the meridional heat transport.*

Unfortunately, the residual circulation in the FESOM1 model can not be easily diagnosed, as also mentioned in the response to reviewer #1. Firstly, this is due to the finite element core of the model, and secondly because no sub-monthly output was stored. We do not think that monthly mean output will be enough to distinguish sufficiently well between high and

low-resolution setups; this is because the high-res diagnostics will lose a significant part that occurs on shorter, sub-monthly timescales.

Since we agree on the high relevance of this diagnostic, for FESOM1's successor, FESOM2, we already handle this differently, and part of the diagnostic is already computed online at runtime (Sidorenko et al. 2020). For future papers, this diagnostic can be included fairly easily.

Having said that, using salinity as a proxy for changes in the residual circulation has also been performed with some success in previous studies, e.g. in Armour et al. (2016), and we thus followed this approach based on the available output.

References:

Sidorenko, D., Danilov, S., Koldunov, N., Scholz, P., and Wang, Q.: Simple algorithms to compute meridional overturning and barotropic streamfunctions on unstructured meshes, *Geosci. Model Dev.*, 13, 3337–3345, <https://doi.org/10.5194/gmd-13-3337-2020>, 2020.

Armour, K. C., Marshall, J., Scott, J. R., Donohoe, A. & Newsom, E. R. Southern Ocean warming delayed by circumpolar upwelling and equatorward transport. *Nat. Geosci.* 9, 549–555 (2016). <https://doi.org/10.1038/ngeo2731>

I think the paper would overall benefit greatly from a more quantitative analysis and statistical significance testing. For example, the differences in the simulations presented in figure 3 and figure 5 do not contain error bounds (e.g. based on natural variability of the control simulations), the differences are not tested for significance, and any of the maps showing changes do not show any significance of the change either.

We have adopted the suggestion of the reviewer and added (non-)significance hatching for the maps/fields shown in Figures 3 and 5. We really like the reviewer's idea to compare to natural variability in the model. That is why the hatching is based on where the simulated changes are smaller than internal variability (two standard deviations) estimated from the control runs. This "physical" significance test reveals that the magnitude of the rather large signals (and the difference in the responses between HR and LR) cannot be explained by natural variability as estimated from the 1950-control simulations (see Methods section "Sea-ice and ocean potential temperature changes compared to unforced variability in the controls", I. 339-349).

6. Related to the issue 5, the difference in sea-ice response in the simulation is very assertively being attributed to the MHT (e.g. lines 155 to 158). However, other processes surface or subsurface heat flux changes, dynamical sea ice changes, etc. might also be critical. The methods section (line 225 to 235) seems to attempt to be more quantitative in this attribution. However, it is not extensively discussed in the main text. I would think that if one would want to formally attribute the sea-ice volume response to a mechanism, it would require a full heat budget analysis that should be discussed in the main text.

We have rewritten the Discussion section. In the revised version we make it clear that one aspect (or one possible reason) for the resolution-dependence lies in the response of the

oceanic MHT in AWI-CM, which is different for the high-resolution (HR) compared to the low-resolution (LR) configuration. To support this, we added a new CMIP5 MHT analysis, where we find a similar tendency for the relatively higher-resolved CMIP5 models compared to the lowest-resolved models.

Furthermore, the Discussion is now more balanced: it lists more of the relevant existing literature, and it also discusses other possible mechanisms that could explain part of the resolution dependence, including those that relate to the mean state (for example, that thermodynamic sea ice processes may be critical, as discussed in the papers led by Cecilia Bitz).

We also state that *“As outlined below, the exact reason for this behavior remains to be determined. However, the faithful representation of ocean eddies by the laws of physics rather than physical parametrizations seems to be one plausible major contributor.”* (l. 200-202)

7. If it was really the difference in MHT causing the different sea-ice response, there should be a similar relation in the CMIP5 models. MHT can easily be obtained from the CMIP5 models. So, I am wondering why the authors do not do a similar analysis as presented in Figure 1 in terms of ocean resolution for the relation between sea-ice change and MHT change? This would certainly provide supporting evidence for their claims if correct.

Following the reviewer’s suggestion, we analysed the response of MHT under climate change (2070-2099 minus 1990-2019) also for the available CMIP5 models (Fig.5d and Supplemental Figure S9). The line plots were, however, not readily available from the CMIP archive; instead, we diagnosed the MHT in the models following the so-called “zig zag” method given in Outten et al. (2018).

In order to perform the CMIP5 MHT analysis, we thus derive global northward heat transport from the CMIP5 heat flux variables "hfx" and "hfy" in x- and y-direction, respectively, thereby accounting for peculiarities of the individual native model grids:

hfx	ocean_heat_x_transport	mon
-----	------------------------	-----

hfy	ocean_heat_y_transport	mon
-----	------------------------	-----

(for example, the MPI-ESM model provides the following data at the German Climate Computing Centre (DKRZ), <https://www.dkrz.de/up/services/data-management/projects-and-cooperations/ipcc-data/cmip5-variables/cmip5-ocean-variable>)

The variables "hfx" and "hfy" were available only for a subset of the models listed in Fig.1, namely, ACCESS1-0, CMCC-CESM, CMCC-CM, CNRM-CM5, GISS-E2-R, IPSL-CM5A-LR,

IPSL-CM5A-MR, IPSL-CM5B-LR, MPI-ESM-LR, MPI-ESM-MR, MRI-CGCM3, NorESM1-M, NorESM1-ME.

Interestingly, when grouping the CMIP5 models into two groups of similar size based on their spatial resolution in the Southern Ocean (7 vs 6 models; resolution being better or worse than a threshold of 90 km) and taking their ensemble-mean, we find the following:

“When it comes to changes in meridional ocean heat transport (MHT), the lower-resolved CMIP5 models show a pronounced southward MHT increase south of 65°S, whereas the relatively higher-resolved CMIP5 models show a weaker increase (between AWI-CM-HR and AWI-CM-LR, Fig. 5d), consistent with our proposed mechanism.” (l. 251-255)

The value of 90km also fits Fig. 1b, dividing the group of “blue” and “red” models that tend to show positive or negative SIE trends over the historical period, respectively. We thus introduce this threshold already early in the revised paper:

“... the models coarser than about 90 km, on the other hand, show negative trends amounting to as much as -0.8 million km^2 per decade (Fig. 1b).” (l. 62/63)

Reference:

Outten, S., Esau, I. & Ottera, O. H. : Bjerknes compensation in the CMIP5 climate models. J. Clim. 31, 8745 – 8760 (2018). <https://doi.org/10.1175/JCLI-D-18-0058.1>

8. I am also wondering why the CMIP5 results are not more strongly incorporated in the manuscript. The discussion in the main text is very brief and the findings are completely absent from the abstract and conclusions. However, I do think that this is strong evidence for a resolution dependence and the discussion should be expanded. Most certainly this analysis requires a methods section and please follow the guidelines for CMIP5 data usage and citation. It would be also useful to include the AWI HR and LR simulations in Figure 1 for reference.

Initially, the CMIP models were used mostly as initial evidence for a resolution dependence. Since they do not reach into the higher resolution regime (even the most highly-resolved CMIP5 model is not eddy-permitting), we then studied our dedicated experiments with AWI-CM where only the ocean resolution is changed, and nothing else was “tuned”.

We thank the reviewer for the assessment that there is strong evidence for a resolution-dependence of Antarctic sea ice trends. As suggested, we added more in-depth analyses of the CMIP models, both in regard to meridional heat transport (Figure 5d and Supplementary Fig. S9) and changes in the ice-edge location (Figure 5e). We identified a resolution-dependence of the MHT response and incorporated this result into the Discussion. There, we now also refer explicitly again to Fig.1, since both reviewers highlighted the importance of this analysis:

“In our study, we also considered results from the wide range of CMIP5 models, which are generally rather coarse, and found additional evidence for a resolution-dependence of the

timing of Antarctic sea-ice decline (Fig. 1). When it comes to changes in meridional ocean heat transport (MHT), the lower-resolved CMIP5 models show a pronounced southward MHT increase south of 65°S, whereas the relatively higher-resolved CMIP5 models show a weaker increase (between AWI-CM-HR and AWI-CM-LR, Fig. 5d), consistent with our proposed mechanism.” (l. 251-255)

Furthermore, the CMIP5 analyses are now explained in detail in a dedicated Methods section (“CMIP5 analyses of sea ice extent, volume, and ocean heat flux”, l. 271-310). We cite Taylor et al. (2012) in the Introduction, and we changed the CMIP citation to the standard citation text in our Acknowledgements section (l. 643-647):

“We acknowledge the World Climate Research Programme’s Working Group on Coupled Modelling, which is responsible for CMIP, and we thank the climate modeling groups for producing and making available their model output. For CMIP, the U.S. Department of Energy’s Program for Climate Model Diagnosis and Intercomparison provides coordinating support and led development of software infrastructure in partnership with the Global Organization for Earth System Science Portals.”

Finally, we included the AWI-CM-LR and HR simulations (for the period 2019-2058) in the revised Figure 1 as suggested. Other periods are discussed in the response to reviewer #1.

9. There is an issue with inconsistency to the study by Turner et al. (2013), who report a negative historical September SIE trend for the MRI-CGCM3 model, whereas here it is the one with the most positive trend. Can you explain this different result? Please provide a methods section on how this analysis is being performed.

The different result can be explained as outlined below. This point has also led us to extend the CMIP5 trend analysis in Figure 1 with all available ensemble members. It ultimately leads to a lower p-value in Figure 1 (0.03 -> 0.01), so that the extended analysis in the end strengthens our finding.

We can confirm that the apparent inconsistency is largely due to the different time period considered (1979-2005 vs 1979-2018), and, most importantly, the fact that 5 ensemble members were averaged in Turner et al. (2013), while we had previously only been using the first available member. We thus decided to add all ensemble members for all the models.

To give more details, a negative trend of -2.3% per decade is given for the MRI-CGCM3 model in Table 1 of Turner et al. (2013) (<https://journals.ametsoc.org/doi/10.1175/JCLI-D-12-00068.1>). The trend is for the period 1979-2005. Using this shorter period, the trend remains slightly positive (+0.13 Mio km² / decade) for the first ensemble member, even when annual-mean SIE extent is considered. We can reproduce the documented value of -2.3% per decade on average when averaging over all five historical ensemble members (which translates to about -0.3 million square km / decade), as was done by Turner et al. (2013).

However, the given trend by Turner et al. (2013) was computed until 2005 only, and we are considering trends until 2018. Since the MRI-CGCM3 model has only one single RCP8.5

ensemble member in the archive and our analysis goes until 2018, nothing changes for that data point in Figure 1. Nevertheless, we have redone the entire analysis in Fig.1 using all ensemble members for those models that do have multiple members for both historical times (until 2005) and for the RCP8.5 extension. This is explained in a new “Methods” section (*CMIP5 analyses of sea ice extent, volume, and ocean heat flux, I. 271-310*), as suggested by the reviewer.

10. In my view, the manuscript suffers from poor or unclear language and formulation (some examples below). I do not have the time to go into this in detail, but I suggest that the language needs to be improved substantially before I would be able to recommend the publication of this manuscript.

We followed all suggested changes mentioned by the reviewer below.

11. Figures: Please use appropriate and consistent font size in all figures, provide all labels and SI units (consistently, e.g. not mixing K and degC) for all figures, and complete x and y axes. Please use a more appropriate color scale for the contour plots (see e.g. <http://www.hclwizard.org/> or <https://doi.org/10.1175/BAMS-D-13-00155.1>). And please follow the journal guidelines.

We use the unit [°C] now consistently throughout the manuscript. We also entirely changed the colormaps in Figures 3 and 4. The colormaps are now a simple Red-Blue colormap for the differences (cm.RdBu in Python), and we used the CMOCEAN package (<https://matplotlib.org/cmoccean/>), which makes sure that the colormaps are color blind-friendly and optimized towards being perceptually uniform. We now apply appropriate color scales for potential temperature (cmoccean.cm.matter) and salinity (cmoccean.cm.dense) in Figure 4.

Particularly in Figure 4, all font sizes, ticks, and labels were strongly increased. The font sizes in Figure 5 are also much more consistent now and we strongly increased tick sizes.

We also removed unnecessary labeling referring to the “HighResMip protocol” that we had overlooked in Figures 3 and 5 before submission.

The guidelines (<https://www.nature.com/documents/ncomms-submission-guide.pdf>) seem to imply that should our manuscript be accepted, we will receive more extensive editorial instructions for final submission of display items that we will of course follow.

Minor issues:

Due to time-constraints, I am only listing a couple of minor issues here, but there is a lot more and I feel that the authors should be overall more careful in terms of wording and presentation and follow guidelines more accurately.

- Abstract 2nd sentence (no line numbers provided): Unspecified “This”: What is “this” referring to?

“This” was changed to “This absence of decline”.

- Abstract: “;” is overused in the abstract and also the text, which makes it much more difficult to read

We changed it to only one occurrence in the abstract. We also deleted several occurrences in the main text.

- Abstract (and later in the text): “a stronger northward branch” is confusing to me. If only the northward branch was stronger but not the southward branch, this would empty out the Southern Ocean in terms of volume. Do you refer to the “upper circulation cell of the [...]”?

We deleted this wording from the Abstract. Later in the text, we tried to be more specific with our wording, e.g. “The heat transport associated with the northward residual branch of the upper circulation cell (black arrow in Fig. 4a), made visible as a prominent minimum salinity layer, causes a reduction of the poleward heat transport around 45°S”.

We added “upper circulation cell” in several other places in the text.

- Abstract: Please tone down “a milestone”, which is in my view not a suitable wording for an abstract.

Deleted. We rephrased to “step” in the revised manuscript.

- Throughout the manuscript there is a lot of claim that things are done “for the first time”. In my view this is unnecessary and should be avoided since provides the reader with an uneasy feeling that the authors are overselling their results.

We certainly do not want to provide this feeling to the reader, since the results certainly show the novelty of our work. We thus deleted some occurrences of statements referring to “things being done for the first time”. However, we keep it in two key statements in the Introduction and Discussion, where we think the statement helps to put our study in context:

Introduction (l. 73-76):

“To our knowledge, besides more idealized climate change simulations (Newsom et al. 2016, Bryan et al. 2014, Goddard et al. 2017) or relatively short integrations (McClean et al. 2011, Kirtman et al. 2012) at eddy-resolving resolutions, this is the first time that comprehensive climate change projections until the end of the century have been performed that involve an ocean model that is regionally eddy-resolving.”

Discussion (l. 195/196):

“In this study, we show for the first time in CMIP-type simulations that a high-resolution climate model is capable of simulating a stable Antarctic sea ice cover over the past few decades --- consistent with available satellite data.”

- Line 5: I don't think reference 5 is most suitable here. The peer-reviewed studies reporting the 2016 sea-ice decline in detail are Turner et al. (2017) and Schlosser et al. (2018).

We removed the reference to the non-peer-reviewed phys.org-article by Arblaster et al. and cite the Turner et al. (2017) and Schlosser et al. (2018) studies.

- Line 6: “some” and “others” needs references

We deleted “some” since there is no peer-reviewed published document stating this explicitly, although we have been part of discussions where the possibility of imminent change was discussed. We thus rephrased to “which one could consider harbingers of imminent change”.

“Others” was referring to the citation at the end of the sentence (Stuecker et al. 2017); we moved the reference to “others” directly.

- Lines 8 to 11: I think one of the most widely accepted and not listed hypothesis is that the historical sea-ice expansion is driven by sea-ice dynamics (Holland and Kwok, 2012; Haumann et al., 2014)

We had listed this important hypothesis (dynamic sea-ice transport changes) in the submitted manuscript in line 9, citing Haumann et al. (2016). However, we did not cite the earlier studies, which we added in the revised version.

- Line 34: What is meant by “not entirely robust in time”?

We deleted this statement. Originally, we meant that the historical period/the satellite era shows a significant dependence on resolution. Towards the end of the century, when sea ice extent in all models decreases similarly strongly, we would not expect to see a clear resolution dependence anymore.

- Figure 1: How is the grey box defined? Please provide an objective measure, i.e. plus-minus two standard deviations or similar.

We thank the reviewer for spotting that the definition was missing in the caption of Figure 1. We used a width around OBS of plus/minus 3 million square kilometres, which is now added to the caption. The value was subjectively chosen as a balance to restrict the analysis to models without obvious very large biases, without being overly restrictive at the same time. Regarding the reviewer’s question on the sensitivity to the chosen box, we did the following:

Including more models at the boundary of the grey box (larger window of +/- 4 million square kilometer), the p-value in Fig.1 becomes lower, which shows that the choice of the box is not critical for our analysis. Wider windows increase the p-value again; however, at some point, the mean contour of the modelled September sea ice extent will be overly shifted meridionally and does not compare at all to the observed September sea ice extent - rendering an explanation for possible common physical processes in the models and reality (which might be different at different latitudes) virtually impossible.

The choice of the grey box and the sensitivity of the p-value to its width is now discussed in the “Methods” section (“*CMIP5 analyses of sea ice extent, volume, and ocean heat flux*”, l. 271-310).

- *Figure 2: It would be clearer to also show a third panel with the difference between the simulations.*

Done. We added difference panels to Figure 2 in polar stereographic projection. These are focused on the Southern Ocean where the largest changes occur between the simulations.

- *Lines 49 to 51: The discussion of the North Atlantic and Arctic seem an unnecessary tangent.*

Although the first reviewer had a similar remark, we would like to keep these few lines to illustrate that in fact we are analyzing a global climate simulation that performs well with respect to other previously discussed phenomena. This is also the reason why we keep global maps in Fig.2.

However, as mentioned above, we added difference panels in polar stereographic projection for the Southern Ocean where the largest changes occur. The remainder of the paper focuses exclusively on the Southern Ocean.

- *Line 58: Please quantify how different the September sea ice extent is rather than writing “quite different”*

We removed “quite” and added a colon as more details are given in the following paragraph. Here, we quantify the sea ice decline in LR and HR:

“The SIE decline in the coarse-resolution LR experiment (-0.49 million km² per decade between 1979--2018; Supplementary Figure S2) is similar to ...” (l. 91/92)

For HR, however, *“the projected HR sea-ice extent shows no clear decline until about the year 2050 (still only about -0.13 million km² per decade between 2019--2058; Fig. 1), and even afterwards the decline progresses with a reduced rate compared to the LR experiment (Fig.3).” (l. 100-102)*

- *Line 64 to 66: Please reformulate the sentence to provide the reader only with relevant information and do not include subjective measures.*

We removed subjective measures and just state facts (l. 97-100): *“Interestingly, there is evidence for multi-decadal sea ice variability from reconstructions (53–55). These suggest a 20th century decline of Antarctic sea ice extent, by up to 20% between the 1950s and 1990s (54), which precedes the later increase found in post-1979 satellite data (55).”*

- *Line 75 and 80: I do not understand what is meant by “physical nature” and “underlying physics”. What is the alternative if it is not the physics?*

As a modeller, one possibility is always that we face a technical issue (e.g., a possible shortness of the spinup). Therefore, we wanted to be extra careful with the wording and exclude technical issues that can arise in high-resolution setups.

- Line 83: I don't think that one can speak of an "ensemble member" when the models configuration changes. An ensemble member is usually run with the same configuration but slightly perturbed initial conditions.

We had the CMIP "variant_label" nomenclature in mind (typically *r1i1p1f1* for the first ensemble member), where experiments can be not just "perturbed initial conditions" (realization $k=1,2,3,4,5$ usually), but there are more choices to generate an ensemble such as "perturbed physics" (m). A perturbed physics ensemble member could be *r1i1p2f1* ($m=2$), for example.

The following text is copied from

https://www.earthsystemcog.org/site_media/projects/wip/CMIP6_global_attributes_filenames_CVs_v6.2.6.pdf :

"variant_label: a label constructed from 4 indices stored as global attributes:

variant_label = r<k>i<l>p<m>f<n> , where

k = realization_index, l = initialization_index, m = physics_index, n = forcing_index"

They could all be considered as different "ensemble members". From that perspective, the very same ocean configuration is run with two different atmosphere resolutions/configurations (different atmospheric physics). We clarified the formulation in this way.

- Line 84: I would disagree that "the delayed sea-ice decline lies solely in the high-resolution ocean". It is difficult to say from ED Fig. 10, since the original configuration is not included and no quantitative measure to support this statement is provided. However, the simulations to look different.

We agree with the reviewer that the simulations look different, too, which is the main message of the figure (now Supplementary Fig. S7). We rephrased the sentence in a more cautious way as follows (l. 117/118):

"The mixed-resolution simulations reveal that the cause for the delayed sea-ice decline is likely to be found in the high-resolution ocean"

The remaining paper then lends more support to our hypothesis that the high-resolution ocean is the cause for the observed differences.

- Line 97: It is not "given" that warmer waters will be entrained with time. They could also just occur locally at the subsurface without affecting the surface.

We deleted that sentence and just kept the facts (l. 130-132): *“Starting with the water mass properties, LR shows a pool of warm subsurface waters southward of 60°S, which is up to 1°C warmer than observed in the upper 500 m (Fig. 4d,f).”*

- Line 106: I do not think that “52” is a suitable reference here. Maybe the article by Marshall and Speer (2012) or similar is more suitable.

Agreed and cited.

- Line 109 and 119: I disagree that the upwelling waters are relatively “cold”. They are actually warmer than the surface. If they were cold, there would be no southward heat transport into the upwelling region.

Yes, the question is relative to what, and compared to the surface the upwelling waters are normally warmer. To reduce confusion, we deleted “cold” from the two mentioned sentences.

- Lines 110 to 112: The latitudinal ranges of upwelling and subduction are not correct. Please refer to literature for more suitable values.

According to Stramma and England (1999), for Antarctic Intermediate Water / Subantarctic Mode Water subduction we changed the value to “north of about 50°S”.

We only found a model study where the authors refer to a broader upwelling range between 40°S and 70°S (Lauderdale et al. 2017); we updated the value accordingly.

References:

Lauderdale, J.M., Williams, R.G., Munday, D.R. et al. The impact of Southern Ocean residual upwelling on atmospheric CO₂ on centennial and millennial timescales. *Clim Dyn* 48, 1611–1631 (2017). <https://doi.org/10.1007/s00382-016-3163-y>

Stramma, L., and England, M. (1999), On the water masses and mean circulation of the South Atlantic Ocean, *J. Geophys. Res.*, 104(C9), 20863–20883, <https://doi.org/10.1029/1999JC900139>

- Line 110: “nutrient-rich” seems out of context

Agreed. We deleted it here and only mention nutrients in the Discussion.

- Line 112 to 114: These two lines seem out of context and distract the reader from the main story. If anything, they might occur in the introduction or conclusion sections.

Agreed. We included this statement in the rewritten Discussion.

- Line 119: I do not see any evidence of realistic formation of deep and bottom waters. Please provide supporting evidence for such a statement.

We refer to previous simulations with the HR ocean grid now cited in the paper (l. 155/156):
“...and supports a more realistic formation of deep and bottom waters, as was shown in previous simulations using the HR ocean grid (Sein et al., 2018).”

Reference:

Sein, D. V. et al.: The relative influence of atmospheric and oceanic model resolution on the circulation of the North Atlantic ocean in a coupled climate model. *J. Adv. Model. Earth Syst.* 10, 2026–2041 (2018). <https://doi.org/10.1029/2018MS001327>

- Line 122: the statement in parenthesis seems out of context.

We deleted the whole statement related to atmospheric cross-equatorial heat transport and changes of the monsoon.

- Line 127: If MHT is generally poleward, why should the upwelling water than be relatively “cold” (line 110 and 119)

Yes, MHT is generally poleward in that latitude range; we do not refer to the relatively “cold” waters (originally lines 110 and 119) anymore, since it is a relative statement that can lead to confusion.

- Line 129: “northward residual branch”: Note the difference between MHT and residual overturning circulation.

To be more precise, we now say “heat transport associated with the northward residual branch of the upper circulation cell” and rephrased as follows (l. 172/173):

“The heat transport associated with the northward residual branch of the upper circulation cell (black arrow in Fig.4a), made visible as a prominent salinity minimum layer, causes a reduction of the poleward heat transport around 45°S.”

- Line 134: Figure 2 does not show sea-surface temperature but 2-m air temperature. Please add a figure of SST or replace figure 2.

We refer now to near-surface temperatures in this line and rephrased as follows: *“The surface and upper subsurface — especially along the sea-ice edge — thus remains considerably cooler than in LR (compare Fig. 5e,f), as clearly seen in projected near-surface temperature changes northward of the ice edge at roughly 60°S (compare Fig. 2c,f).” (l. 177-179)*

- Lines 144-153: Note the difference between MHT and residual overturning circulation. Please use precise and accurate formulations. Also plot the residual overturning circulation and the differences to provide supporting evidence. “entirely unclear” is not correct, since there is existing literature (see major issues above). Note that eddy saturation is not relevant to this discussion (see e.g. Rintoul, 2018).

“Unclear” is a literal quote from the paper by Poulsen et al. (2018), but we agree that with the additional literature suggested by the reviewer this statement should be toned down. We deleted “entirely unclear” from the text, just refer to eddy compensation, and rephrased as follows with the reviewer’s comments in mind (l. 212-225):

“One possible reason for the resolution dependence lies in the different response of the oceanic meridional heat transport (MHT) in AWI-CM for high-resolution (HR) compared to low-resolution (LR). This may be due to differences in the representation of the underlying eddy-induced circulation and the degree to which it can oppose or compensate the changes in the wind-driven overturning. A wind-driven increase of the upper clockwise meridional circulation cell, with the northward branch being its upper representation, may be partly balanced by a compensating counter-clockwise eddy overturning in LR, similar to partial (parameterized) eddy compensation as shown for CMIP5 models (Downes et al. 2013). In HR, the compensating counter-clockwise eddy overturning seems to increase more slowly than the wind-driven increase. In line with this, the projected anomalous MHT for HR associated with the upper circulation cell shows an increased northward heat transport across 60°S at the end of the century when compared to LR (Fig.5d). This is despite similar increases of 7% and 5% in peak zonal-mean wind stress in HR and LR, respectively (Supplementary Fig. S8). Because of the lack of centennial projections at high resolution (Poulsen et al. 2018), it has not been directly shown yet how eddy compensation will evolve in high-resolution climate projections, i.e. over longer time scales that go beyond the historical period. Our results thus provide further insights into this issue; and they highlight its importance for explaining structural uncertainties associated with Antarctic sea-ice projections and the Antarctic sea ice paradox.”

- Line 157: I don't think that it could be claimed that “this difference alone is sufficient”. There would need to be supporting evidence from the entire CMIP5 ensemble.

This sentence is not present anymore in our rewritten Discussion (although we found supporting evidence that similar MHT changes occur in the CMIP5 models when grouped based on their spatial resolution, see answer above).

- Lines 155-189: Please avoid colloquial language.

We have rewritten the Discussion.

- Line 186: The effect of “better-resolved bathymetry” is not explained anywhere in this manuscript and I am not sure how this relates here.

We had added this point because it will be of considerable interest in the future. Higher resolution is not only about better resolving meso-scale features, it also comes along with a better-resolved bathymetry (if observational data is available). It has been shown that regional features strongly impact heat fluxes towards the Antarctic continental shelf, e.g. localized transports related to canyons (e.g. Morrison et al., 2020). We however decided to delete the sentence from the text.

- Lines 188-189: *I don't think that such far reaching claims can be made. Please reformulate.*

The link of ice shelf basal melting to sea ice cover is given in Hellmer et al. (2012, <https://doi.org/10.1038/nature11064>). In their study, they show that projected loss of sea ice in the Weddell Sea could lead to more warm water of open ocean origin reaching far beneath the Filchner-Ronne Ice Shelf. This is what we had in mind when writing the last sentence.

We also moved the discussion of impacts of the underlying residual overturning on biogeochemistry (which was included in Results before) to the Discussion.

Altogether, we rephrased the last sentences in the paper as follows (l. 263-269):

"The implications of this go beyond the physical climate system as "measured" by the Antarctic sea ice cover. In fact, the upwelled waters around Antarctica have considerable nutrient concentrations and sustain up to 40% of the low-latitude primary production (Primeau et al. 2013, Hauck et al. 2018). Future changes in the amount of carbon and nutrients being transported along this pathway will impact global productivity and air-sea CO₂ flux substantially (Moore et al. 2018, Hauck et al. 2018). A better representation of the Southern Ocean can thus have major implications for biogeochemistry and the global carbon cycle, for the uptake of anthropogenic heat by the ocean and hence global mean temperature change, and for the trustworthiness of representing basal melting of Antarctic ice-shelves (Hellmer et al. 2012) and in consequence global sea level rise."

- Lines 236-240: *Please revise the Data availability statement to comprise with journal guidelines. I don't think "on reasonable request" is in line with the guidelines.*

The wording "on reasonable request" was taken from the journal's website directly (<https://www.springernature.com/gp/authors/research-data-policy/data-availability-statement/s/12330880>). We contacted the editor who is handling this paper, and she confirmed that at a minimum, a statement confirming that all relevant data are available from the authors *on reasonable request* is sufficient, which at this stage we adopted here given the immense amount of high-res model data.

Should the paper be accepted, we plan to make some of the data (or at least the scripts to generate figures) available in a publicly accessible repository.

- Line 279: *I don't think that this is an appropriate (peer-reviewed) reference here.*

We certainly agree and removed the "Arblaster et al. (2019)" reference to the phys.org-website. In fact, the reference had been suggested in a previous round of reviews.

References:

- Bitz, C. M., and Polvani, L. M. (2012), Antarctic climate response to stratospheric ozone depletion in a fine resolution ocean climate model, *Geophys. Res. Lett.*, 39, L20705, doi:10.1029/2012GL053393.
- Fyfe, J.C., O.A. Saenko, K. Zickfeld, M. Eby, and A.J. Weaver, 2007: The Role of Poleward-Intensifying Winds on Southern Ocean Warming. *J. Climate*, 20, 5391–5400, <https://doi.org/10.1175/2007JCLI1764.1>
- Griffies, S.M., M. Winton, W.G. Anderson, R. Benson, T.L. Delworth, C.O. Dufour, J.P. Dunne, P. Goddard, A.K. Morrison, A. Rosati, A.T. Wittenberg, J. Yin, and R. Zhang, 2015: Impacts on Ocean Heat from Transient Mesoscale Eddies in a Hierarchy of Climate Models. *J. Climate*, 28, 952–977, <https://doi.org/10.1175/JCLI-D-14-00353.1>
- Haumann, F. A., Notz, D., and Schmidt, H. (2014), Anthropogenic influence on recent circulation-driven Antarctic sea ice changes, *Geophys. Res. Lett.*, 41, 8429– 8437, doi:10.1002/2014GL061659.
- Holland, P., Kwok, R. Wind-driven trends in Antarctic sea-ice drift. *Nature Geosci* 5, 872–875 (2012). <https://doi.org/10.1038/ngeo1627>
- Marshall, J., Speer, K. Closure of the meridional overturning circulation through Southern Ocean upwelling. *Nature Geosci* 5, 171–180 (2012). <https://doi.org/10.1038/ngeo1391>
- Meredith, M. P., and A. M.Hogg (2006), Circumpolar response of Southern Ocean eddy activity to a change in the Southern Annular Mode, *Geophys. Res.Lett.*, 33, L16608, doi:10.1029/2006GL026499.
- Meredith, M.P., A.C. Naveira Garabato, A.M. Hogg, and R. Farneti, 2012: Sensitivity of the Overturning Circulation in the Southern Ocean to Decadal Changes in Wind Forcing. *J. Climate*, 25, 99–110, <https://doi.org/10.1175/2011JCLI4204.1>
- Morrison, A. K., O. A. Saenko, A. McC. Hogg, and P. Spence (2013), The role of vertical eddy flux in Southern Ocean heat uptake, *Geophys. Res.Lett.*, 40, 5445–5450, doi:10.1002/2013GL057706
- Rintoul, S.R. The global influence of localized dynamics in the Southern Ocean. *Nature* 558, 209–218 (2018). <https://doi.org/10.1038/s41586-018-0182-3>
- Schlosser, E., Haumann, F. A., and Raphael, M. N.: Atmospheric influences on the anomalous 2016 Antarctic sea ice decay, *The Cryosphere*, 12, 1103–1119, <https://doi.org/10.5194/tc-12-1103-2018>, 2018.
- Screen, J.A., N.P. Gillett, D.P. Stevens, G.J. Marshall, and H.K. Roscoe, 2009: The Role of Eddies in the Southern Ocean Temperature Response to the Southern Annular Mode. *J. Climate*, 22, 806–818, <https://doi.org/10.1175/2008JCLI2416.1>

Turner, J., T.J. Bracegirdle, T. Phillips, G.J. Marshall, and J.S. Hosking, 2013: An Initial Assessment of Antarctic Sea Ice Extent in the CMIP5 Models. *J. Climate*, 26, 1473–1484, <https://doi.org/10.1175/JCLI-D-12-00068.1>

Turner, J., Phillips, T., Marshall, G. J., Hosking, J. S., Pope, J. O., Bracegirdle, T. J., and Deb, P. (2017), Unprecedented springtime retreat of Antarctic sea ice in 2016, *Geophys. Res. Lett.*, 44, 6868– 6875, doi:10.1002/2017GL073656.

REVIEWER COMMENTS

Reviewer #1 (Remarks to the Author):

I congratulate the authors for their work to improve this manuscript during a difficult year. I am largely happy with the responses to my queries and appreciate the effort that has been put into this. However, I think there are a couple of issues which still need further consideration before I could recommend acceptance.

1. The emphasis on the residual overturning. I asked in my review that the residual overturning be calculated explicitly rather than inferred from salinity (Lines 23, 145-156, 172-173, 214-220). I totally understand that this may not be possible given the model output (as described in the responses). However, I wonder if the level of speculation on this point should be reduced in the discussion?

2. I think that the resolution of the simulations should be compared with the Rossby radius and Hallberg (2013) referenced (and the Rossby radius marked on figure 1b)? The second reviewer raised the question about whether these simulations were eddy resolving (or Eddy Rich as termed by Griffies et al, 2015). On line 70, the resolution is described as being as high as 8km over most of the Southern Ocean region but when I read the methods(line 289) I see that the resolution is ~14km in the 45-65S latitude band. If that is the case, then the resolution is close to ORCA025 coupled simulations with NEMO (HadGEM3 and CNRM) which is 11km at 65S and 20km at 45S. The CCSM3.5 results were from a 1/10 degree model which would be termed eddy rich. I think putting into context with the Rossby radius would give clarity on whether the HR simulations are eddy rich or eddy present resolution. Even if eddy present, these are still cutting edge simulations for projecting 21st century change but it would raise questions on the role of eddies. In particular, note in Roberts et al. (2019) that the eddy present ORCA025 resolution has a very weak ACC and this is improved by either lower or higher resolution.

Other comments:

1. I don't think it is correct to say that the sea ice has not declined without qualifying this with on average or mentioning a few years of decline (lines 12 and 31)
2. I cannot see why 90 km should be a separation point (line 62). This would seem to separate what is described as ~1 degree models from coarser models-is that correct? If so, perhaps discussion in the original 1 degree coupled models of Gordon et al (2000) or Gent et al (1998) might be helpful?
3. Line 203-205 What about GEOMETRIC as described in Mak et al., 2018, JPO?

Minor comments:

1. L15 Wording is awkward (better 'projected 21st century sea ice changes')
2. L21 should read not projected to decline until mid 21st century
3. L45 near term doesn't seem right-I read that as the next decade whereas I think over this century is intended?
4. Line 81-83 I still think that this detail is unnecessary and I would redraw figure 2a,b, d,e as polar stereographic for Antarctica (global maps could be in supplementary but are not sufficiently relevant here)
5. Line 94 the decadal variability here in HR is also evident in HadGEM3-MM (Roberts et al., 2019) with the ORCA025 ocean (which I believe is a similar resolution at these latitudes)
6. Line 137 I don't understand the 'joint functioning'
7. Line 183 should this read Equatorward?
8. Figure 5 is too busy-this looks like 3 figures condensed into one. I think this needs to have less panels-maybe remove the sea ice fraction?
9. Line 188 this needs discussion of eddy saturation (see papers by Munday et al)
10. What is meant by 'modern formulation' on line 228

Reviewer #3 (Remarks to the Author):

Review comments for "Delayed Antarctic sea-ice decline in high-resolution climate change simulation" by Rackow et al. (NCOMMS-20-03460A).

This is my first review of this paper. I am an additional reviewer to be asked to check whether the authors' responses to reviewer #2 are reasonable. In my reading the manuscript, reviewer comments, and the response, it seems that there is one major issue in this paper. Although the authors assumed that the HR simulation is eddy-resolving (e.g., L73-76), it is not valid. Judging from Figs 4b and 4c in Sein (2016), the HR in this study is an eddy-permitting model for the Southern Ocean (as the original reviewer2 pointed out). The authors' replies to the reviewers' comments are reasonable, except reviewer #2-4. As long as the ocean model in this paper is eddy-permitting over the Southern Ocean, the authors can not obtain the conclusions in eddy-resolving models. I suggest that the authors put some comments about the possibility raised by reviewer #2 (comment 4) in the Discussion. It is OK to revise/update your own results when the author can perform a true eddy-resolving model.

Better sea-ice representation in an eddy-permitting model is useful for the Antarctic and Southern Ocean research communities, and it is valuable for publishing in Nature Communications. Therefore, please state clearly that HR is an eddy-permitting model in the manuscript.

Here are some comments in my reading.

I read the manuscript, assuming that the eddy-permitting model can represent the meanders of ACC and other ocean currents. If so, please describe the roles of mesoscale dynamical processes (meander/standing eddy and small transient eddies) in the early stage of the manuscript. Furthermore, some figures showing ocean flows (vectors, barotropic stream-function, or surface stream-function) in the two different resolutions are helpful for readers to see the difference in the model representations.

L87-89: The sentence is unnecessary in the context.

Figure 4: I didn't understand why the 30W section was used. It seems to me that zonal-mean properties are suitable, following Armour et al. (2016).

Response to reviewers

We would like to express our gratitude to the reviewers for their thorough reviews. Following their comments and recommendations, we have revised the paper as follows.

1. We added a discussion on the local Rossby radius to our paper, following Hallberg (2013), and included it in Figures 1, 2, and S5, as suggested by reviewer #1.
2. We added more discussion on the comment by the original reviewer #2 that the HR resolution is mostly eddy-permitting. This still turns out to be “cutting edge” over

latitudes of the ACC, as commented by reviewer #1, where even a global regular $1/10^\circ$ model is not eddy-resolving anymore.

Supplementary Figure S4: Snapshots of the Southern Ocean surface circulation [m/s] for September 1990 in the two different AWI-CM ocean configurations.

a, The high-resolution HR grid, **b**, the low-resolution LR grid. Note the logarithmic scale of the panels. The Antarctic Circumpolar Current is represented with eddy-permitting resolution or better in HR when compared to the local Rossby radius of deformation (see Figure S5).

3. We added new figures with a snapshot of the Southern Ocean surface circulation (see above Figure S4) and of the barotropic streamfunctions in AWI-CM-HR and -LR (Figure S12). S4 illustrates the very different representation of the eddy fields visually, while S12 confirms that AWI-CM-HR does not suffer from a weak ACC, since by design its resolution at these latitudes is better than in the widely used ORCA025 0.25° configuration (Roberts et al., 2019; Supplementary Figure S5).

Attached to this letter you find our detailed point-by-point answers to the remaining reviewers' comments. We have also uploaded a difference file (diff_main_revised2.pdf) with added/removed text compared to the previous version in blue and red, respectively.

Best regards, Thomas Rackow / *On behalf of all co-authors*

Reviewer #1 (Remarks to the Author):

I congratulate the authors for their work to improve this manuscript during a difficult year. I am largely happy with the responses to my queries and appreciate the effort that has been put into this. However, I think there are a couple of issues which still need further consideration before I could recommend acceptance.

We thank the reviewer very much for this positive assessment and the kind words. In the following, we provide point-by-point answers to the remaining issues.

1. The emphasis on the residual overturning. I asked in my review that the residual overturning be calculated explicitly rather than inferred from salinity (Lines 23, 145-156, 172-173, 214-220). I totally understand that this may not be possible given the model output (as described in the responses). However, I wonder if the level of speculation on this point should be reduced in the discussion?

We thank the reviewer for the understanding. In the last paragraph of the discussion, we removed the single mentioning of “residual overturning” and just refer to “circulation” now. We did the same for the abstract, where only differences in the meridional heat transport are mentioned now (which we show explicitly). There is only one more paragraph (l.237-l.250) with some speculation on the residual overturning. In order to reduce this here as well, we changed the following sentences:

L. 242-244: “In HR, the compensating counter-clockwise eddy overturning seems to increase more slowly than the wind-driven part, *although this will need to be shown more directly in future studies.*”

(added the last part to emphasize our speculation)

L. 247-250: “Because of the lack of centennial projections at high resolution⁷⁷, it has not been directly shown yet how eddy compensation will evolve in high-resolution climate projections, i.e., over longer time scales that go beyond the historical period. Our results thus provide further insights into this issue; and *they suggest a role for explaining structural uncertainties associated with Antarctic sea-ice projections and the Antarctic sea-ice paradox.*”

(We removed “[the results] highlight its importance for explaining structural uncertainties” since it is not explicitly shown here yet. However, we think it is justifiable to suggest at least a link to eddy compensation from what we can show - and from the discussion of previous literature.)

2. I think that the resolution of the simulations should be compared with the Rossby radius and Hallberg (2013) referenced (and the Rossby radius marked on figure 1b)?

Done. We added the Rossby radius to Figure 1b, with a vertical dashed line for its mean

value and a grey box denoting its range in the considered latitudinal band. This shows that AWI-CM-HR is an eddy-permitting model and in some areas locally eddy-resolving. We also updated a similar plot in the supplement (Supplementary Figure S3).

The second reviewer raised the question about whether these simulations were eddy resolving (or Eddy Rich as termed by Griffies et al, 2015). On line 70, the resolution is described as being as high as 8km over most of the Southern Ocean region but when I read the methods (line 289) I see that the resolution is ~14km in the 45-65S latitude band. If that is the case, then the resolution is close to ORCA025 coupled simulations with NEMO (HadGEM3 and CNRM) which is 11km at 65S and 20km at 45S. The CCSM3.5 results were from a 1/10 degree model which would be termed eddy rich. I think putting into context with the Rossby radius would give clarity on whether the HR simulations are eddy rich or eddy present resolution.

Thank you for this very relevant comment. We agree that the previous text could have been interpreted as saying that the resolution is 8 km everywhere in the Southern Ocean, which is not the case. Our high-resolution unstructured mesh, which is designed to be at least

eddy-permitting (and in some parts eddy-resolving) over the core of the ACC, is difficult to describe just with words (like in the Methods) or simple measures like the average resolution in a band (because the ACC core “wiggles” in and out of this band). That is why we now added additional panels to Figure 2 that show the HR and LR mesh resolution in multiples of the Rossby radius of deformation R , as suggested by the reviewer. Typically, when the grid spacing is $<0.5R$, this is termed “eddy-resolving” according to Hallberg (2013), and the regime between $0.5R$ and $1R$ is termed “eddy-permitting”.

Supplementary Figure S5: Grid spacing of different ocean model grids as multiples of the local Rossby radius of deformation. *a*, ORCA025 NEMO (0.25°), *b*, a typical 0.1° ocean grid, *c*, AWI-CM-HR, and *d*, AWI-CM-LR.

Using the strict definition above, the figure shows that the HR model grid is, by design, eddy-permitting over the whole core of the ACC by focusing resolution here, and it goes down to eddy-resolving resolution in parts of the Southern Ocean (Brazil-Malvinas and Agulhas regions, as is now written in the Methods):

L. 342-345: *“One set of simulations (AWI-CM-HR) was run at eddy-permitting ocean resolution over the ACC region with some further eddy-resolving refinements to about 8 km over the Agulhas and Brazil-Malvinas Confluence regions (see map in Fig. 4b,c by Sein et al.37).”*

It is very important to note that even an expensive global $1/10^\circ$ model like CCSM3.5, which is “eddy-rich” over mid-latitudes, is not eddy-resolving over the whole core of the ACC. Similarly, the ORCA025 NEMO grid is typically termed “eddy-permitting” (in mid-latitudes), but one can see clearly that it is not eddy-permitting over the whole ACC, in particular not in the Pacific and Indian Ocean sectors. From this analysis, we think that it is fair to say that the HR grid used in our AWI-CM projections is closer to what a $1/10^\circ$ model achieves (in terms of resolution at latitudes of the ACC) than to the ORCA025 configuration, so it is difficult to label the HR grid.

To make all these points clearer, we changed the main text as follows in several places and also cite Hallberg (2013):

L. 76-85: *“AWI-CM employs a sea ice-ocean model formulated on multi-resolution unstructured meshes⁴¹. This technique permits ocean eddies over the whole core of the ACC when compared to the local Rossby radius of deformation² by concentrating grid points in that area (Fig.2g,i). Over the ACC, the HR grid is thus comparable in resolution to a $1/10^\circ$ ocean model that is typically termed “eddy-rich” over mid-latitudes, but stops being eddy-resolving in polar regions² (Supplementary Fig. S4 and S5). [...] To our knowledge, besides more idealized climate change simulations^{32,33,43} or relatively short integrations^{44,45} at eddy-rich resolutions, this is the first time that comprehensive climate change projections until the end of the century have been performed using an ocean model that is eddy-permitting in the vicinity of the whole ACC core.”*

We also added the definitions following Hallberg (2013) to the Methods section in lines 342-347 (resolution $r < \text{Rossby radius } R$: eddy-permitting; $r < 0.5R$: eddy-resolving).

Even if eddy present, these are still cutting edge simulations for projecting 21st century change but it would raise questions on the role of eddies. In particular, note in Roberts et al. (2019) that the eddy present ORCA025 resolution has a very weak ACC and this is improved by either lower or higher resolution.

With the discussion above and the fair comparison even to 0.1° models in the Southern Ocean over the ACC region, we think that the role of eddies can be discussed similarly well with AWI-CM-HR as in previous studies using global 0.1° models. In particular, the resolution in HR is better over the ACC than in the ORCA025 grid, which is not eddy-permitting throughout all of the ACC (Supplementary Fig. S5). Compared to the ORCA025 configuration in Roberts et al. (2019), the ACC transport through Drake Passage is 178Sv with the HR ocean grid (see answer to reviewer #3 below with a plot of the barotropic streamfunction). HR thus matches the recent observational range of 173 ± 11 Sv (Donohue et al., 2016; Roberts et al., 2019) very well, while LR with its 133Sv shows lower transport, but still well within the CMIP5 range of 155 ± 55 Sv (Meijers et al., 2012). Both configurations do

not suffer from a very weak ACC as seen by Roberts et al. (2019), which could indeed be related to the fact that our AWI-CM simulations are either lower resolved (LR, where the eddy parameterization is active) or higher resolved (HR) than the ORCA025 setup. We have added a new paragraph with this information to the Discussion section (l. 223-236).

References:

Roberts, M. J., Baker, A., Blockley, E. W., Calvert, D., Coward, A., Hewitt, H. T. et al.: Description of the resolution hierarchy of the global coupled HadGEM3-GC3.1 model as used in CMIP6 HighResMIP experiments, *Geosci. Model Dev.*, 12, 4999–5028, <https://doi.org/10.5194/gmd-12-4999-2019>, 2019.

Donohue, K. A., Tracey, K. L., Watts, D. R., Chidichimo, M. P., and Chereskin, T. K.: Mean Antarctic Circumpolar Current transport measured in Drake Passage, *Geophys. Res. Lett.*, 43, 11760–11767, <https://doi.org/10.1002/2016gl070319>, 2016.

Meijers, A. J. S., Shuckburgh, E., Bruneau, N., Sallee, J.-B., Bracegirdle, T. J., and Wang, Z.: Representation of the Antarctic Circumpolar Current in the CMIP5 climate models and future changes under warming scenarios, *J. Geophys. Res.-Oceans*, 117, C12008, <https://doi.org/10.1029/2012JC008412>

Other comments:

1. *I don't think it is correct to say that the sea ice has not declined without qualifying this with on average or mentioning a few years of decline (lines 12 and 31)*

We agree that we should be more specific in the beginning, since we mention the few years of decline for the first time only in lines 35-37. We followed your suggestion and added “on average” to line 13, and “multi-decadal decline” in the following sentence:

L. 13-16: *“Despite ongoing global warming and strong sea-ice loss in the Arctic, on average the observed Antarctic sea-ice extent has not declined since 1979 when satellite data became available. This absence of multi-decadal decline is in stark contrast to existing climate change simulations that tend to show a strong negative sea-ice trend for the same period.”*

Similarly for lines 32-34:

“In contrast, satellite data show a much more stable sea-ice cover on average, with a slight (albeit statistically non-significant) multi-decadal positive trend of +0.11 million km² per decade in September for the same period (1979--2018)”

We also mention “few years of decline” now in lines 35-36, and added a reference to Eayrs et al. (2021) who discuss this “naturally occurring variability”. In their closing paragraph, Eayrs et al. also state that after the rapid decline in 2016, for now “it seems that Antarctic sea-ice cover has returned to the average climatological conditions of the modern satellite era”, so we also cite it for this view:

L. 35-38: *“In the few years from 2016 onwards, there has been significantly lower Antarctic sea-ice extent³⁻⁸, which one could consider harbingers of imminent change. However, others⁹ have argued that sea-ice extent is expected to regress to the near-neutral decadal trend in the near future; and indeed data for 2020/2021 seem to support this view⁸.”*

Added Reference No. 8:

Eayrs, C., Li, X., Raphael, M.N. et al. Rapid decline in Antarctic sea ice in recent years hints at future change. *Nat. Geosci.* 14, 460–464 (2021).

<https://doi.org/10.1038/s41561-021-00768-3>

2. I cannot see why 90 km should be a separation point (line 62). This would seem to separate what is described as ~1 degree models from coarser models-is that correct? If so, perhaps discussion in the original 1 degree coupled models of Gordon et al (2000) or Gent et al (1998) might be helpful?

We changed “the models coarser than about 90km” back to “*the coarsest models*” (l.67), since we agree that one cannot identify a clear separation point from this plot.

3. Line 203-205 What about GEOMETRIC as described in Mak et al., 2018, JPO?
(<https://doi.org/10.1175/JPO-D-18-0017.1>)

For context, lines 203-205 have been: *“The impact of ocean resolution has been discussed in previous modelling studies (...) and some of the results, in particular the sign of the MHT response, may seem contradictory to what is concluded here. However, as detailed below, a meaningful comparison with previous studies is surprisingly difficult to carry out given major differences...”*

We are not convinced that the study by Mak et al. (2018) is closely related here, since we are talking mainly about the sign of the response in MHT and the ambiguity introduced by GM. GEOMETRIC is another scheme to compute the GM coefficient, based on an energy consideration. Even though such a scheme for the GM coefficient might be more physically based than the parameterization used by us, it remains a parameterization that requires substantial testing in realistic setups.

We recognize and agree that there is potential in using more sophisticated parameterizations for GM. Understandably, we have not tried all of them in our studies (e.g., Greatbatch and Eden also proposed a parameterization for the GM coefficient that relies on eddy energy balance), because each requires long tests also in realistic setups, and it remains to be seen whether more sophisticated parameterizations for the GM coefficient provide more consistency between coarse and high-resolution simulations.

Minor comments:

1. L15 Wording is awkward (better ‘projected 21st century sea ice changes’)

Thank you for this suggestion, done!

2. L21 should read *not projected to decline until mid 21st century*

Done.

3. L45 *near term doesn't seem right-I read that as the next decade whereas I think over this century is intended?*

Yes, we mean “until the end of this century” and thus deleted “near-term”.

4. Line 81-83 *I still think that this detail is unnecessary and I would redraw figure 2a,b, d,e as polar stereographic for Antarctica (global maps could be in supplementary but are not sufficiently relevant here)*

We changed the panels to polar stereographic projection and moved the global maps to the Supplement (Figure S6), as suggested by the reviewer. We kept lines 81-83 to be able to refer from the main text to the new Figure S6 with global maps (now lines 88-92).

5. Line 94 *the decadal variability here in HR is also evident in HadGEM3-MM (Roberts et al., 2019) with the ORCA025 ocean (which I believe is a similar resolution at these latitudes)*

We have added the citation as follows (L. 100-102):

“The high-resolution experiment HR, on the other hand, exhibits enhanced decadal-scale variability in sea-ice extent between 1950--1990 (...), which has been found in other high-resolution models (Hewitt et al. 2016; Roberts et al. 2019).”

As discussed above, the new Supplemental Figure S5 shows the differences in resolution between ORCA025 and AWI-CM-HR.

6. Line 137 *I don't understand the 'joint functioning'*

We mean the combined effect of GM and Redi diffusivity. We therefore introduce GM now in this line and rephrased as follows:

L.144-146: *“Another contribution could be the combined effect of the Gent--McWilliams (GM) parameterization (Gent and McWilliams, 1990) for meso-scale eddies, which acts to flatten isopycnals, and isoneutral (Redi) diffusivity.”*

7. Line 183 *should this read Equatorward?*

Southwards of 60°S (max. at about 65°S), LR shows a poleward (southward) MHT anomaly while HR shows an equatorward (northward) MHT anomaly in Fig. 5d. We thus think that our sentence is correct:

“In contrast to HR, our low-resolution climate change experiment shows increased poleward MHT even south of 60°S (Fig. 5d).”

8. *Figure 5 is too busy-this looks like 3 figures condensed into one. I think this needs to have less panels-maybe remove the sea ice fraction?*

We followed the reviewer's suggestion from round 1 of reviews (to add the sea-ice fraction to the original panels 5b and 5d), and we still think this was a very good idea. Furthermore, an entirely new CMIP5 analysis on the location of the September sea-ice edge in all CMIP5 models was added for this purpose as well (see Methods 1.327-332, and Fig.5e). This info would be lost by removing the sea-ice fraction again and reverting to the original Figure 5. However, we still have the original Figure 5 without the sea-ice fraction info and can use it for the final version should the editor/graphics team also agree on this.

9. *Line 188 this needs discussion of eddy saturation (see papers by Munday et al)*

L. 188 has been in context: *"Bryan et al. explain this difference with increased (approximately doubled) heat transport by standing eddies, the meanders of the ACC, which are more pronounced in the high-resolution configuration. The heat flux response by transient eddies can thus be much smaller in HR than in LR (where only transient eddy effects are parameterized), despite similar increases in westerly winds."*

We now mention the concept of eddy saturation explicitly before. As a response to wind changes (e.g., due to a CO₂ doubling), the eddy field will be such that it compensates for the change in forcing, resulting in a remarkably insensitive ACC transport. This could be through a change in transient eddies and/or through a change in meandering flows. We added the following sentences before the original L.188 with a citation of Munday et al. 2013:

L. 191-193: *"According to the concept of eddy saturation (Munday et al. 2013), ocean eddies compensate for changes in external forcing. Both transient and standing eddies may contribute to this phenomenon, and the simulated compensation is dependent on parameterizations in low-resolution models."*

Although it is relevant also for eddy compensation/saturation, we want to add that the change in the structure of meanders is not the subject of GM, in other words entirely out of the scope of what parameterizations can achieve, and we mention this in lines 258-261:

"[...] Bryan et al. (2014) also highlight a role for standing eddies, i.e. meanders of the ACC, in balancing the wind-driven overturning. This mechanism of ACC equilibration will need further investigation in future studies, as its effects are fully beyond the scope of what eddy parameterizations in current climate models can achieve."

References:

Munday, D. R., Johnson, H. L., & Marshall, D. P. (2013). Eddy Saturation of Equilibrated Circumpolar Currents, Journal of Physical Oceanography, 43(3), 507-532.
<https://doi.org/10.1175/JPO-D-12-095.1>

10. What is meant by ‘modern formulation’ on line 228

We mean it in the sense of the cited paper by Bryan et al. 2014, who refer as “modern” to models that

“use a variable formulation for the GM coefficient, and not a constant coefficient, in order to get the correct response in the ACC region to changes in zonal wind stress (...). Further refinement of how the GM coefficient is specified as a function of ocean model variables might help reduce the differences seen in the LR and HR responses to the transient forcing of increasing CO₂ documented in this work.”

Current CMIP models show varying degrees of GM sophistication in that sense, using e.g. constant coefficients only. Bryan et al. also state earlier that:

“The GM coefficient is now a function of space and time following the implementation in Danabasoglu and Marshall (2007) and decreases with depth varying as the square of the local buoyancy frequency.”

The GM parameterization in AWI-CM is modern in this sense, meaning that the reference diffusivity is not constant but scaled in the horizontal dimension by the local resolution. As detailed by Wang et al. (2014), tapering functions following Danabasoglu and McWilliams (1995) and Large et al. (1997) are also applied. There is also a scaling of the GM coefficient in the vertical dimension with depth following the buoyancy frequency N^2 , making the GM coefficient smaller with depth (following an implementation by NCAR). However, one could think of new approaches that specify GM as a function of ocean model variables, or flow-dependent, as mentioned by Bryan et al. (2014). We have thus rephrased to:

L. 251-253: *“It can be argued that either dedicated tuning or new flow-dependent approaches for how the space and time-dependent GM coefficient is determined in modern eddy parameterizations³² of climate models with lower-resolution ocean grids could overcome some of the shortcomings discussed in this study.”*

Reviewer #3 (Remarks to the Author):

Review comments for "Delayed Antarctic sea-ice decline in high-resolution climate change simulations" by Rackow et al. (NCOMMS-20-03460A).

This is my first review of this paper. I am an additional reviewer to be asked to check whether the authors' responses to reviewer #2 are reasonable.

We thank the reviewer very much for the careful reading of the manuscript. In the following, we give point-by-point answers to the identified issue.

In my reading the manuscript, reviewer comments, and the response, it seems that there is one major issue in this paper. Although the authors assumed that the HR simulation is eddy-resolving (e.g., L73-76), it is not valid. Judging from Figs 4b and 4c in Sein (2016), the HR in this study is an eddy-permitting model for the Southern Ocean (as the original reviewer2 pointed out).

We thank the reviewer for pointing out this apparent issue, which can indeed be confusing. However, due to the finite-element numerical core of the ocean model, the model is indeed eddy-resolving (albeit locally) in some areas of the Southern Ocean, as is written in line 74, although being mostly eddy-permitting over the ACC (also l. 74). We have added panels g)-i) to Figure 2 to make this clearer and changed all occurrences of “permitting”/”resolving” in the main text accordingly.

As also written in the response to reviewer #1, a regular $1/10^\circ$ model that is generally termed “eddy-rich” (because it is eddy-rich over mid-latitudes) is only “eddy-permitting” over large parts of the ACC (Supplementary Fig. S5). Therefore, the resolution of our HR grid is actually comparable at these latitudes and “cutting-edge”, as stated by reviewer #1 above. This is in line with Hallberg (2013), who states that “one should ask *where*, not *whether*, a global ocean model can explicitly represent eddies”. This study is now cited and we think our additional statements put our study in much better context to the previous resolution studies (e.g. L.76-80 and L. 223-236).

References:

Hallberg, R. (2013): *Using a resolution function to regulate parameterizations of oceanic mesoscale eddy effects*. Ocean Modelling, Volume 72, December 2013, Pages 92-103.

The authors' replies to the reviewers' comments are reasonable, except reviewer #2-4. As long as the ocean model in this paper is eddy-permitting over the Southern Ocean, the authors can not obtain the conclusions in eddy-resolving models. I suggest that the authors put some comments about the possibility raised by reviewer #2 (comment 4) in the Discussion. It is OK to revise/update your own results when the author can perform a true eddy-resolving model.

As written above, our extended analysis shows that also a $1/10^\circ$ “eddy-resolving” ocean model (over mid-latitudes) is only eddy-permitting over parts of the ACC core in the Southern Ocean, and one needs to be very precise about which region can explicitly represent ocean eddies. The AWI-CM-HR configuration comes close to $1/10^\circ$ over the core of the ACC in terms of resolution, but none of these configurations is truly eddy-resolving in the Southern Ocean yet. However, we think that we found a way to write this very carefully at several places in the main text.

Reviewer #2's comment #4 from the previous round was: “Given the points 1-3 above, I am worried that some of these differences to the published literature might result from the fact that the here presented high-resolution (HR) model is not fully eddy resolving, but rather

eddy permitting and that at the same time the eddy parametrization is switched off. So, I am concerned that if one would go to even higher resolutions than HR the results would look more like what is being published in literature.”

We thank the reviewer for the suggestion to add a comment on the possibility raised by reviewer #2. In terms of ACC transport, as mentioned by reviewer #1 above, Roberts et al. (2019) have shown that a 0.25° model without eddy parameterization can indeed be in some kind of “grey zone”, meaning that either lower-resolved models (with a parameterization) or higher-resolved models (without parameterization) immediately improve the simulated ACC transport. In that regard, the AWI-CM-HR simulation does not seem to be in that regime -despite not using an eddy parameterization in the Southern Ocean- and shows a realistic transport, likely because it is already more highly resolved:

L. 225-232: “The same holds for “eddy-permitting” configurations like the widely used ORCA02552 at 0.25° resolution, which is eddy-permitting only over parts of the ACC (Supplemental Fig. S5). This can coincide with a very weak ACC transport through Drake Passage on the order of 90 Sv and was shown to improve at either lower or higher resolution⁵². Similar to a 1/10° model at high latitudes, the AWI-CM-HR configuration discussed in this paper is by construction eddy-permitting over the entirety of the ACC core (Supplemental Fig. S5). In terms of barotropic transport around Antarctica (Supplemental Fig. S12), AWI-CM-HR matches the recent observational Drake Passage transport range of 173±11 Sv^{52, 73} well (178 Sv), while AWI-CM-LR with its 133 Sv shows lower transport, but still well within the CMIP5 range of 155±55 Sv.”

However, we also updated the Discussion as follows by adding the possibility raised by the original reviewer #2 that aspects of the simulations might be different as soon as truly eddy-resolving models in high latitudes are used:

L. 234-236: “Fully eddy-resolving models, with local resolutions better than 3-4 km over the ACC, are currently under development, and it remains to be seen how these models will perform in the Southern Ocean.”

Better sea-ice representation in an eddy-permitting model is useful for the Antarctic and Southern Ocean research communities, and it is valuable for publishing in Nature Communications. Therefore, please state clearly that HR is an eddy-permitting model in the manuscript.

We thank the reviewer very much for this assessment. We now clearly state the resolution of HR (see also Figure S4):

*L. 74-78: “Here we analyse experiments with an **eddy-permitting** and locally eddy-resolving global model system (AWI-CM-HR)³⁶⁻³⁹ (down to 8 km resolution **in some parts** of the Southern Ocean, see Methods) that has been integrated until the end of the 21st century, following the HighResMIP protocol⁴⁰. AWI-CM employs a sea ice-ocean model formulated on multi-resolution unstructured meshes⁴¹. **This technique permits ocean eddies over the***

whole core of the ACC when compared to the local Rossby radius of deformation⁴² by concentrating grid points in that area

We also removed the reference to “eddying” from the abstract (which might sound like an “eddy-rich” model); we compare it to other model configurations like 0.25° and 0.1° in Figure S5, and we also added figures that put the resolution into context with the local Rossby radius of deformation in a quantitative way (Figure 2 and Supplementary Figure S5).

Here are some comments in my reading.

I read the manuscript, assuming that the eddy-permitting model can represent the meanders of ACC and other ocean currents. If so, please describe the roles of mesoscale dynamical processes (meander/standing eddy and small transient eddies) in the early stage of the manuscript. Furthermore, some figures showing ocean flows (vectors, barotropic stream-function, or surface stream-function) in the two different resolutions are helpful for readers to see the difference in the model representations.

We thank the reviewer for these suggestions. We changed a paragraph early in the manuscript accordingly (in the introduction):

L. 48-55: “It has been suggested that models with a faithful representation of meso-scale ocean eddies in the Antarctic Circumpolar Current (ACC) are needed in order to better represent the behaviour of the Southern Ocean²⁷. In fact, realistically representing the Southern Ocean circulation and its response to global warming is known to be a challenging task in coarse-resolution models²⁸, requiring parameterizations (e.g. for the effect of ocean eddies) that are sensitive to how they are tuned. In particular, changes in forcing can be compensated via eddy compensation and saturation²⁹, resulting in a notably insensitive transport of the ACC through Drake Passage. Both changes in meso-scale transient eddies and changes in meandering flows (standing eddies) may contribute to these phenomena, and the degree to which eddy compensation and saturation is realised in simulations thus depends on parameterizations in low-resolution models.”

To support this visually, as suggested by the reviewer we added a new Figure S4 of the surface circulation (speed in m/s, see first figure in this response) to the Supplement, which gives a clear intuitive impression of the degree of realism in the HR simulations as well as the different representation of meandering flows compared to LR.

Furthermore, we added a more objective figure where the different resolutions are compared to the local Rossby radius of deformation (Figure 2g,h,i); and another Figure S12 with the different barotropic streamfunctions in the 1950 controls was added to the Supplement (see below). This information should allow the reader to get a better picture of the differences between the different resolutions.

Supplementary Figure S12: Barotropic streamfunction in the Southern Ocean for the different ocean configurations in the 1950-control simulations. a, AWI-CM-LR, b, AWI-CM-HR. The ACC transport through Drake Passage (value at Cape Horn) is 133 Sv (LR) and 178 Sv (HR), respectively. To calculate the barotropic streamfunction, the barotropic flow is integrated northwards starting from the Antarctic coast.

In Figure S12, one can see that HR matches well the recent observational Drake Passage transport range of 173 ± 11 Sv (as given by Donohue et al., 2016 and by Roberts et al., 2019, a comparison suggested by reviewer #1), while LR lies in the CMIP5 range of 155 ± 55 Sv (Meijers et al., 2012) with its 133 Sv.

References:

Donohue, K. A., Tracey, K. L., Watts, D. R., Chidichimo, M. P., and Chereskin, T. K.: Mean Antarctic Circumpolar Current transport measured in Drake Passage, *Geophys. Res. Lett.*, 43, 11760–11767, <https://doi.org/10.1002/2016gl070319>, 2016.

Meijers, A. J. S., Shuckburgh, E., Bruneau, N., Sallee, J.-B., Bracegirdle, T. J., and Wang, Z.: Representation of the Antarctic Circumpolar Current in the CMIP5 climate models and future changes under warming scenarios, *J. Geophys. Res.-Oceans*, 117, C12008, <https://doi.org/10.1029/2012JC008412>

L87-89: The sentence is unnecessary in the context.

We moved this sentence to the discussion section, where we already mention the potential impacts on representing Antarctic ice shelf melting and thus also Antarctic Ice Sheet behavior in Earth System models:

L. 293-295: “A reduced warming in high vs low-resolution models at depth could also have strong implications for current efforts to estimate the basal melting of Antarctic ice shelves⁹¹ and for projected Antarctic Ice Sheet behavior in models.”

Figure 4: I didn't understand why the 30W section was used. It seems to me that zonal-mean properties are suitable, following Armour et al. (2016).

We used the 30°W section for illustration because the water mass structure with the “sandwiching” of low-salinity Antarctic Intermediate Water is nicely seen here. Oceanographers are more familiar with this structure than a zonal mean and recognize the known pattern. In addition, it is a well-visited section through the Weddell Sea because of many nearby campaigns that happened over time (see image below).

Image taken from WOCE ATLAS: http://woceatlas.tamu.edu/printed/SOA_CRUISE.html

The illustration will be muted when taking the zonal mean because the real Southern Ocean is not at all zonally symmetric. However, for the more relevant Figure 5 that mimics Armour et al.'s analysis, we have indeed taken a zonal mean to be very consistent with their analysis and in order not to miss potentially different behavior of the other ocean basins. Also, our Meridional Heat Transport (MHT) analysis in Figure 5 includes zonal-mean quantities in order not to miss any meridional transport, as was also done by Armour et al. (2016).

REVIEWERS' COMMENTS

Reviewer #1 (Remarks to the Author):

Following my second review of this manuscript, I have read the revised manuscript and the authors responses to review comments. I am satisfied with both the responses and the revised manuscript with one request. My request is that in addition to reference 27 (or instead of), chapter 9 of AR6 is referenced (Fox-Kemper et al.) as this is the source of the assessment on low confidence in Antarctic sea ice projections.

Response to reviewers

We would like to express our gratitude to the reviewers for their reviews. Following their request, we have revised the paper as follows. We have also uploaded a difference file (diff_main_final.pdf) with added/removed text compared to the previous version in **blue** and **red**, respectively. Note that we have followed an Editorial Author Checklist, which explains the other changes to the Abstract (shortening to max. 150 words) and to the Introduction (restructuring).

Reviewer #1 (Remarks to the Author):

Following my second review of this manuscript, I have read the revised manuscript and the authors responses to review comments. I am satisfied with both the responses and the revised manuscript with one request. My request is that in addition to reference 27 (or instead of), chapter 9 of AR6 is referenced (Fox-Kemper et al.) as this is the source of the assessment on low confidence in Antarctic sea ice projections.

We thank the reviewer very much for this positive assessment and all previous reviews. We agree and have added the reference to WG1's Chapter 9 as a new reference 28 in the final manuscript.

Best regards, Thomas Rackow / *On behalf of all co-authors*